# Equilibration towards generalized Gibbs ensembles in non-interacting theories

**Marek Gluza[1], Jens Eisert[1,2,3] and Terry Farrelly[4⋆]**

**1** Dahlem Center for Complex Quantum Systems,
Freie Universität Berlin, 14195 Berlin, Germany
**2** Department of Mathematics and Computer Science,
Freie Universität Berlin, 14195 Berlin, Germany
**3** Helmholtz-Zentrum Berlin für Materialien und Energie, 14109 Berlin, Germany
**4** Institut für Theoretische Physik, Leibniz Universität Hannover, 30167 Hannover, Germany

⋆ farreltc@tcd.ie

## Abstract

Even after almost a century, the foundations of quantum statistical mechanics are still not completely understood. In this work, we provide a precise account on these foundations for a class of systems of paradigmatic importance that appear frequently as mean-field models in condensed matter physics, namely non-interacting lattice models of fermions (with straightforward extension to bosons). We demonstrate that already the translation invariance of the Hamiltonian governing the dynamics and a finite correlation length of the possibly non-Gaussian initial state provide sufficient structure to make mathematically precise statements about the equilibration of the system towards a generalized Gibbs ensemble, even for highly non-translation invariant initial states far from ground states of non-interacting models. Whenever these are given, the system will equilibrate rapidly according to a power-law in time as long as there are no long-wavelength dislocations in the initial second moments that would render the system resilient to relaxation. Our proof technique is rooted in the machinery of Kusmin-Landau bounds. Subsequently, we numerically illustrate our analytical findings by discussing quench scenarios with an initial state corresponding to an Anderson insulator observing power-law equilibration. We discuss the implications of the results for the understanding of current quantum simulators, both in how one can understand the behaviour of equilibration in time, as well as concerning perspectives for realizing distinct instances of generalized Gibbs ensembles in optical lattice-based architectures.

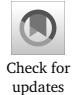
# 1   Introduction

Over more than a century, it has become clear that the methods of statistical mechanics work incredibly well in a vast range of physical situations. But, to this day, a complete understanding of *why* this is the case remains elusive. Based on both experimental and theoretical work, a good deal of progress has already been made [1–6]. Nevertheless, the key objective, finding a set of physical assumptions from which we can demonstrate that quantum systems reach thermal equilibrium, has yet to be achieved. And there are exceptional cases where this simply does not occur, which typically involve the existence of locally conserved quantities.

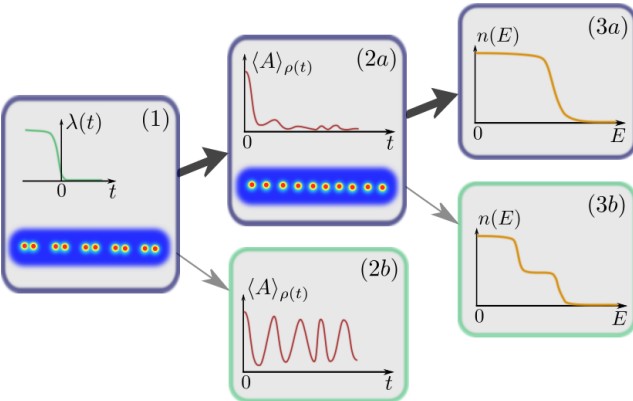

Figure 1: Thermalization and equilibration are often studied in a dynamical quench scenario, where a parameter in the Hamiltonian is suddenly quenched to zero, which knocks the system out of equilibrium (1). The subsequent process of (generalized) thermalization has two components. First, the system must relax to a steady state (2a) with respect to meaningful quantities. Exceptions to this are typically characterized by oscillations, as in (2b). Second, if equilibration occurs, the equilibrium state must be thermal (exemplified here by the Fermi-Dirac distribution in (3a)), or correspond to a generalized Gibbs ensemble (3b) in case further constants of motion are relevant.

The process whereby a system locally relaxes to a thermal state or a generalized Gibbs ensemble (which we call generalized thermalization) can be broken down into two components (see Fig. 1). The first is simply that it equilibrates, meaning the system spends most of the time locally close to some time-independent steady state. This should be true at least for large classes of important observables, e.g., local observables. A crucial aspect (sometimes overlooked) is that the equilibration time for this must be realistic: in experiments, we can observe physical systems relaxing over reasonable times only, which is something that needs to be appreciated. The second component in the case of thermalization is that the equilibrium steady state has no detailed memory of the initial state (beyond, e.g., temperature or chemical potential), namely it is a thermal state.

It has become clear, however, that some specific classes of physical systems do not equilibrate [7–10], at least over the times one can assess in the laboratory. Furthermore, some systems equilibrate but not to a thermal state, instead retaining some memory of the initial state [11–13]. A major distinction arises in this context between non-integrable systems, which indeed are expected to equilibrate to a thermal state, and integrable systems, which are expected not to fully thermalize, but to equilibrate to generalized Gibbs ensembles [14–20]. Many-body localized systems [11,21], in which disorder and interactions interplay in a subtle

manner, can be seen as being reminiscent of the latter systems, as instances of quantum systems which also do not thermalize. In both cases, local (or quasi-local) conserved quantities play a major role. Whenever initial states with inequivalent values of these conserved quantities are experimentally accessible, the resulting steady states will retain a memory of these differences that can be measured. A rigorous *dynamical derivation* of generalized thermalization must therefore overcome several difficulties arising from these observations: we must identify what properties most physical systems have that lead them to thermalize or relax to a generalized Gibbs ensemble.

There are several different theoretical approaches to this challenging long-standing problem. One is to focus on what can be proven for abstract quantum systems with as few assumptions as possible [4, 22–24]. In this case, powerful results have been found, though often without reproducing the relevant equilibration times [25–28]. Another approach is to use randomness to attack the problem [29–34]. Suggestions for the mechanism underlying the relatively fast process of equilibration in the general setting have been offered [35, 36], but a consensus together with more concrete estimates for equilibration times have yet to emerge.

A second approach is to build the analysis on specific physical settings (e.g., the Bose-Hubbard model in the free superfluid regime). But even here there is a dearth of results justifying why the observed times are so short in comparison to the general bounds. Some exceptions in specific cases are, amongst others, presented in Refs. [14, 37, 38]. In particular, studying quenches has been particularly rewarding [39]. In this context, numerical studies often provide useful insights [1–4, 40–48].

In this work, we first analyse quenches of lattice fermions (and – less explicitly – bosons) to non-interacing Hamiltonians. Our first main result is that they locally equilibrate quickly. Two tools we employ are the Kusmin-Landau bound [49] and fermionic Gaussification from Ref. [50]. The latter showed that non-interacting fermions on a lattice locally Gaussify, meaning the state becomes locally indistinguishable from a Gaussian state for relatively long times. However, this Gaussian state may be time dependent. Not only do we show that one of the assumptions of Ref. [50] is unnecessary for Gaussification, but we also show that the Gaussian state that the system approaches will be time independent. This is a proof of equilibration over realistic times for these models, and it also proves that the equilibrium state can be described by a *generalized Gibbs ensemble (GGE)*.

In fact, our work can be seen as a comprehensive rigorous proof of a convergence to generalized Gibbs ensembles, bringing the program initiated in Refs. [14, 19, 51, 52] to an end, by widely generalizing the previous results, while keeping the discussion fully rigorous. We then turn to discussing the question of whether one does indeed need the extra degrees of freedom of a GGE (as opposed to simply a thermal state). We show numerically that initial states corresponding to thermal states of an Anderson insulator equilibrate after quenching the on-site disorder to a thermal state (or grand canonical state), except in cases with highly correlated noise. In this latter case, the equilibrium state must be described by a GGE. It is easy to see that if one has strongly inhomogeneous initial conditions, the equilibration times can be of the order of the system size, see, e.g., Ref. [38]. Finally, we consider some possibilities for realizing distinct instances of generalized Gibbs ensembles in optical lattices and systematically studying their stability in the presence of interactions.

## 2 Sufficient conditions for local equilibration to a generalized Gibbs ensemble

### 2.1 Notions of equilibration

A quantum system locally equilibrates if, for all times $t$ between some relaxation time $t_0$ and some recurrence time $t_R$, the state at time $t$ is practically indistinguishable from the time-averaged state $\hat{\varrho}^{(eq)}$ with respect to local observables [4]. In other words, the extent of non-equilibrium fluctuations is bounded by some small $\epsilon > 0$ such that for every local observable $\hat{A}$ we have

$$|\langle \hat{A} \rangle_{\hat{\varrho}(t)} - \langle \hat{A} \rangle_{\hat{\varrho}^{(eq)}}| \leq \epsilon \tag{1}$$

for all $t \in [t_0, t_R]$, where $\langle \hat{A} \rangle_{\hat{\varrho}} = \text{tr}[\hat{\varrho}\hat{A}]$. Clearly, whenever a system equilibrates, the equilibrium state must be the infinite time average

$$\hat{\varrho}^{(eq)} = \lim_{T \to \infty} \frac{1}{T} \int_0^T \text{dt}\, \hat{\varrho}(\text{t}) \,. \tag{2}$$

While it is highly plausible that systems equilibrate, it is significantly more challenging to identify the equilibration time $t_0 > 0$. When equilibration does indeed occur, a most natural question is how to precisely characterize this equilibrium state. Statistical physics is built upon the assumption that systems equilibrate to a thermal state. The thermal (or Gibbs) state of a quantum system with Hamiltonian $\hat{H}$ is defined to be

$$\hat{\varrho}^{(\beta,\mu)} = \frac{e^{-\beta(\hat{H}-\mu\hat{N})}}{\text{tr}[e^{-\beta(\hat{H}-\mu\hat{N})}]} \,, \tag{3}$$

where $\beta > 0$ is the inverse temperature, which fixes the value of the expected energy, $\mu$ is the chemical potential, which determines the expected particle number, and $\hat{N}$ is the particle number operator. We say that a system with initial state $\hat{\varrho}$ *thermalizes* locally if during the evolution generated by $\hat{H}$ it equilibrates in the sense defined above and if $\hat{\varrho}^{(eq)}$ is locally indistinguishable from the thermal state of $\hat{H}$ (for some value of $\beta$ and $\mu$). For the case of non-interacting, quasi-free models, thermal states of quadratic Hamiltonians are called Gaussian or quasi-free and are the target equilibrium ensemble upon quenches to quasi-free dynamics.

### 2.2 Statement of the main result

Our main result is the following. Take a system of non-interacting fermions on a line described by a translation invariant (with periodic boundary conditions) short-ranged Hamiltonian. Assume that the couplings are generic such that there are no points with coinciding roots of the derivatives $E''(p) = E'''(p) = 0$ of the dispersion relation $E$. Initialize the system in a state with finite correlation length and non-resilient second moments (defined presently). Then local equilibration occurs according to the following statement.

**Theorem 1** (Emergence of statistical mechanics). *There exist a constant relaxation time $t_0$ and a recurrence time $t_R$ proportional to the system size such that for all times $t \in [t_0, t_R]$ the system locally equilibrates to a Gaussian generalized Gibbs ensemble, with*

$$|\langle \hat{A} \rangle_{\hat{\varrho}(t)} - \langle \hat{A} \rangle_{\hat{\varrho}^{(eq)}}| \leq C t^{-\gamma} \tag{4}$$

*for some $C, \gamma > 0$ independent of the system size. That is, we can set $\epsilon = C t_0^{-\gamma}$ in Eq. (1) for $t_0 \leq t \leq t_R$.*

The equilibrium ensemble $\hat{\varrho}^{(eq)}$ is a generalized Gibbs ensemble. Moreover, it is parametrized by an *intensive* number of generalized temperatures that scales with the correlation length $\xi$ of the initial state and the thermodynamical potentials involved are exclusively local. These are two defining features of statistical mechanics and indeed are present in our equilibrium ensemble. We argue this by invoking the Jaynes' principle of looking for the maximum entropy state given expectation values of quantities of interest. In our case, these are the tunnelling currents $\hat{I}_z$ (defined in detail below) which are quadratic operators, e.g., $\hat{I}_0$ is the mean on-site particle density and $\hat{I}_1$ corresponds to the nearest-neighbour tunnelling. Since the equilibrium ensemble is Gaussian we can also use the property that characterizes these states, namely that they are the maximum entropy states given fixed second moments [50]. Hence fixing the values $\langle \hat{I}_z \rangle$ is a way of specifying a Gaussian state. Say that $\epsilon = C t_0^{-1/6}$ is our desired experimental resolution, and deviations from equilibrium should not be larger than this number. Then within that precision we neglect all the currents with range significantly above the correlation length $z > z_\xi \approx \xi \ln(\epsilon^{-1})$ and aim at reproducing in the equilibrium ensemble $\hat{\varrho}^{(eq)}$ the values of the relevant conserved quantities obtained from the initial state

$$I_z = \langle \hat{I}_z \rangle_{\hat{\varrho}(0)} = \langle \hat{I}_z \rangle_{\hat{\varrho}^{(eq)}} \tag{5}$$

for $z \leq z_\xi$. This condition is met by setting the state to be parametrized as

$$\hat{\varrho}^{(eq)} = Z^{-1} e^{-\sum_{z=0}^{z_\xi} \lambda_z \hat{I}_z} , \tag{6}$$

where $Z > 0$ ensures normalization and $\lambda_z$ are Lagrange multipliers. Note that for fixed $\epsilon > 0$, e.g., determined by the experimental resolution of the apparatus, only an *intensive* number of generalized temperatures $\lambda_z$ significantly contributes to the parametrization of this ensemble. It remains to argue that for $z \geq z_\xi$ all correlation functions are smaller than the desired resolution $\epsilon$. By the result in Ref. [53], any one-dimensional thermal state of the type (6) has exponentially decaying correlations with a correlation length bounded by some $\xi_A$. Hence indeed we recover asymptotically $\langle \hat{I}_z \rangle_{\hat{\varrho}_G^{(eq)}} \sim C_{\text{Clust}} e^{-z/\xi_A} \ll \epsilon$. Here we can identify the chemical potential as $\mu = \lambda_0$ and oftentimes $\beta = \lambda_1$, e.g., in the case of the nearest-neighbour hopping quench Hamiltonian. If we find that $\sum_{z=0}^{z_\xi} \lambda_z \hat{I}_z = \beta \hat{H} + \mu \hat{N}$ where $\hat{H}$ is exactly the quench Hamiltonian and $\hat{N}$ the particle number of operator then we would say that the equilibrium ensemble is *thermal*. Whenever this is not the case then one concludes that relaxation towards a *generalized Gibbs ensemble* (GGE) has taken place.

## 2.3 Discussion of the main result

The novel feature beyond known non-interacting results [14, 19, 42, 50–52, 54–57] is that for the first time we show equilibration over a reasonable time in a closed quantum system to occur *generically* within a *class* of models and initial states. Such ubiquitous validity is one of the defining features of statistical mechanics. Roughly speaking, in our case it occurs as a result of translation-invariance of the dynamics, even if the initial state is non-Gaussian and is not translation invariant, as long as it does not have unnatural initial correlations. Note that our argument does without the knowledge of the actual values of the couplings or specific initial configurations of the particles as long as these satisfy our general assumptions. This generality is a crucial feature of statistical mechanics and is to a large degree responsible for its success.

Throughout the work, it will be our goal to give intuition that grounds the proof of this result. Let us begin by explaining how equilibration can fail or is physically implausible if any of the ingredients of Theorem 1 is relaxed and therefore other assumptions become necessary. By our result equilibration occurs via dynamics generated by non-interacting Hamiltonians: while

strong results are possible even in the general interacting case [24,25,28,38,58–60], deriving a rigorous bound on the equilibration time of the type $\epsilon = O(t^{-\gamma})$ has been elusive so far. In fact, it may be impossible on grounds of quantum computational complexity [61–64] because equilibrated time-evolution is concomitant to converging results of a quantum algorithm and often the runtime should be longer than polynomial [65].

In the main text, we will present the results for non-interacting fermions even though same statements hold for non-interacting bosons with a little technical fine-print due to the local Hilbert space being unbounded, and one needs additional assumptions on the correlations in Ref. [51]. Concerning geometry, we consider a ring configuration, mostly for the clarity of the argument while of course thermodynamics should not change by the choice of boundary conditions. However, in higher dimensions additional complications could occur as the group velocity, i.e., the derivative of the dispersion relation could vanish along curves instead of separated points [66], but certainly our techniques should generalize when supplemented with additional assumptions that exclude such technical issues. One of the core physical assumptions enabling sufficient scrambling of the initial conditions is translation invariance of the Hamiltonian. Relaxing it, one can find that particles do not propagate and without mixing ergodicity breaks down and with it relaxation. As a prime example, the Anderson insulator model [67] is a non-translation invariant Hamiltonian where equilibration is obstructed due to localization.

Long-ranged non-interacting models can actually violate causality [68,69]. That is to say, if equilibration occurs, then one would need to develop an entirely new intuition for its mechanisms. Here, we assume a short-ranged local Hamiltonian which is already enough to ensure effective causality by means of the Lieb-Robinson bound [70–74]. By additional technical calculation, it should be possible to extend the results to couplings that asymptotically decay exponentially. Note that we consider a closed system described by a static Hamiltonian. If we relax the condition on exponentially decaying correlations then one can consider as the initial state a state evolved backwards to extensively long times which suddenly would acquire "out of nowhere" non-equilibrium dynamics while the system should be expected to be equilibrated.

Finally, it has turned out to be necessary to demand that second-moments of the fermionic state be non-resilient. The simplest example of a state without this property occurs when particles occupy half of the system and the other half is empty. Then for any short-ranged Hamiltonian by the Lieb-Robinson bound it will take extensive times for the particles to even explore the system and equilibration to occur. This property will be precisely stated below in the form of a definition after the necessary notation has been introduced. Summarizing this discussion, trying to establish equilibration one can encounter numerous obstructions, some of them are fundamental difficulties and some are rather technical. In this work we identify precise conditions, mostly concerning locality of couplings and correlations, which are physically very natural and general, and at the same time are sufficient to establish local equilibration with time-scales for a closed quantum system.

## 3 Class of physical systems considered

### 3.1 Non-interacting fermionic models

We denote fermionic annihilation operators by $\hat{f}_x$ and will discuss bosons in the appendix. The annihilation operators obey the canonical anti-commutation relations $\{\hat{f}_x, \hat{f}_y^\dagger\} = \hat{f}_x \hat{f}_y^\dagger + \hat{f}_y^\dagger \hat{f}_x = \delta_{x,y}$. Note that any fermionic initial state satisfies the parity superselection rule [75,76], meaning physical states can never involve a superposition of even and odd numbers of fermions. More precisely, we assume that the density operator $\hat{\varrho}$ commutes

with $(-1)^{\hat{N}}$, where $\hat{N} = \sum_{x=1}^{L} \hat{N}_x$ is the total number operator with $\hat{N}_x = \hat{f}_x^\dagger \hat{f}_x$.

A non-interacting fermionic model conserving particle number is characterized by a quadratic Hamiltonian of the form

$$\hat{H}(h) = \sum_{x,y=1}^{L} h_{x,y} \hat{f}_x^\dagger \hat{f}_y \,, \tag{7}$$

where $h = h^\dagger \in \mathbb{C}^{L \times L}$ is the coupling matrix for a finite system size $L$. By a linear transformation of the fermionic operators preserving the anti-commutation relations, any such Hamiltonian can be brought into diagonal form. Whenever the system is translation invariant then $h$ is *circulant*, and so $h$ can be diagonalized by a discrete Fourier transform. Throughout, we make the assumption that $h \in \mathbb{R}^{L \times L}$ is real, translation invariant and has range $R$, that is $J_z := h_{1,1+z}$ vanishes for $z > R$ and hence we consider the hopping models of the form

$$\hat{H}(h) = J_0 + \sum_{z=1}^{R} J_z \sum_{x=1}^{L} \hat{f}_x^\dagger \hat{f}_{x+z} + \text{h.c.} \,. \tag{8}$$

By this, we can define the dispersion relation $E : \mathbb{R} \to \mathbb{R}$ as

$$E(p) = J_0 + 2 \sum_{z=1}^{R} J_z \cos(pz) \tag{9}$$

and evaluating at $p_k = 2\pi k/L$ we can write the eigenvalues of $h$ as $\omega_k = E(p_k)$ for any finite system size $L > 2R$. Here $E(p)$ is analytic and its derivative can be used to express the dispersion gaps, e.g., $\omega_{k+1} - \omega_k = E'(\tilde{p}_k)2\pi/L$ for some $\tilde{p}_k \in [p_k, p_{k+1}]$ by the mean value theorem. It will be useful to define $J_{\max} = \max_{z=1,\dots,R} |J_z|$. The Heisenberg evolution of mode operators reads

$$\hat{f}_x(t) = e^{it\hat{H}(h)} \hat{f}_x e^{-it\hat{H}(h)} = \sum_{y=1}^{L} G_{x,y}^*(t) \hat{f}_y \,, \tag{10}$$

where $G^*(t) = e^{-ith}$ is the *propagator* given by

$$G_{x,y}(t) = \frac{1}{L} \sum_{k=1}^{L} e^{i\omega_k t + 2\pi i k(x-y)/L} \tag{11}$$

in the translation invariant case, see Appendix A. The *covariance matrix* is defined as the collection of second moments of a state $\hat{\varrho}$, given by

$$\Gamma_{x,y} = \langle \hat{f}_x^\dagger \hat{f}_y \rangle_{\hat{\varrho}} \,. \tag{12}$$

Observe that physically only the operator $\hat{\Gamma}_{x,y} = \hat{f}_x^\dagger \hat{f}_y$ is not Hermitian and hence not an observable. However, its real and imaginary parts defined as $2\text{Re}[\hat{\Gamma}_{x,y}] = \hat{f}_x^\dagger \hat{f}_y + \hat{f}_y^\dagger \hat{f}_x$ and $2\text{Im}[\hat{\Gamma}_{x,y}] = -i(\hat{f}_x^\dagger \hat{f}_y - \hat{f}_y^\dagger \hat{f}_x)$ are physical observables. Hence, their expectation values can be measured individually in a physical system and then one obtains

$$\Gamma_{x,y} = \frac{1}{2} \langle \hat{f}_x^\dagger \hat{f}_y + \hat{f}_y^\dagger \hat{f}_x \rangle_{\hat{\varrho}} + \frac{i}{2} \langle -i(\hat{f}_x^\dagger \hat{f}_y - \hat{f}_y^\dagger \hat{f}_x) \rangle_{\hat{\varrho}} \,. \tag{13}$$

Note that we consider states with no pairing correlations: $\langle \hat{f}_x^\dagger \hat{f}_y^\dagger + \text{h.c} \rangle = 0$. Our methods can be generalized to that case as well [77, 78], but this complicates the presentation. Using (10) we see that the covariance matrix at time $t$ is

$$\Gamma(t) = G(t)\Gamma G(t)^\dagger \,. \tag{14}$$

Of particular relevance for us will be fermionic Gaussian states, which are completely specified by their second moments and Wick's theorem for higher-order correlation functions [77].

To prove many of our results later, we will require that the initial state has exponential decay of correlations, meaning there exist positive constants $C_{\text{Clust}}, \xi > 0$ such that correlations decay like

$$|\langle \hat{A}\hat{B}\rangle_{\hat{\varrho}} - \langle \hat{A}\rangle_{\hat{\varrho}}\langle \hat{B}\rangle_{\hat{\varrho}}| \leq s(\hat{A})s(\hat{B})C_{\text{Clust}}e^{-d/\xi} \,, \tag{15}$$

where $\hat{A}$ and $\hat{B}$ are observables acting non-trivially only on lattice regions separated by a distance $d$ with sizes $s(\hat{A})$ and $s(\hat{B})$ respectively. For simplicity, we have chosen $\|\hat{A}\| = \|\hat{B}\| = 1$, where $\|\cdot\|$ is the operator norm.

## 3.2 Constants of motion

What are the relevant constants of motion for translation invariant dynamics? The most obvious candidate consists of momentum occupation numbers

$$\hat{n}_k = \frac{1}{L}\sum_{x,y=1}^{L} e^{2\pi ik(y-x)/L}\hat{f}_x^\dagger \hat{f}_y \,. \tag{16}$$

Another set of conserved quantities are the *current* operators

$$\hat{I}_z(\eta) = \frac{1}{L}\sum_{x=1}^{L} e^{i\eta}\hat{f}_x^\dagger \hat{f}_{x+z} + \text{h.c.} \,, \tag{17}$$

where $\eta$ can in sometimes be interpreted as coming from a magnetic field via Peierls substitution. These are indeed conserved quantities, which follows because they are linear combinations of the momentum occupation numbers. The following two extreme cases are important $\hat{I}_z(\eta = 0) = (2/L)\sum_{k=1}^{L}\cos(2\pi kz/L)\hat{n}_k$, cf. e.g. [79] and for $\hat{I}_z(\eta = \pi/2) = -(2/L)\sum_{k=1}^{L}\sin(2\pi kz/L)\hat{n}_k$. For the latter type of currents to be *present* it is necessary that the covariance matrix as defined above is not real.

The current operators allow us to judge how many conserved quantities are really necessary to describe the steady state with finite experimental resolution $\epsilon$. Due to the exponential decay of correlations Eq. (15), we have $|\langle \hat{I}_z \rangle| \leq C_{\text{Clust}}e^{-z/\xi}$, and so $|\langle \hat{I}_z \rangle| \leq \epsilon$ for $z \geq \xi \ln(C_{\text{Clust}}/\epsilon)$. So there are only $z \sim \xi$ non-negligible values of $\langle \hat{I}_z \rangle$ which constitute the only relevant local conserved quantities. Thus, whenever equilibration occurs, then the equilibrium ensembles of any set of non-local momentum occupation numbers $\{\langle \hat{n}_k \rangle\}$ with the same current content will agree.

For initial states $\hat{\varrho}(0)$ with short range correlations we prove in the appendix, assuming minimal degeneracy of the dispersion relation $\omega_k$, that the steady-state obtained from the infinite-time average $\Gamma_{x,y}^{(\infty)}$ is translation invariant up to a small parameter

$$\left|\Gamma_{x,y}^{(\infty)} - \Gamma_{x,y}^{(\text{eq})}\right| \leq C_I L^{-1} \,, \tag{18}$$

where $C_I$ is independent of the system size. We can define the equilibrium values by a real-space *average*

$$\Gamma_{x,y}^{(\text{eq})} = \frac{1}{L}\sum_{z=1}^{L}\Gamma_{x+z,y+z} \,. \tag{19}$$

We then can find the Peierls angle by setting $\eta_z = \arg[\Gamma_{1,z}^{(\text{eq})}]$. By this, we find that our target equilibrium ensemble has matrix elements which agree with the *initial* expectation value of the conserved operator

$$I_{|x-y|} = \Gamma_{x,y}^{(\text{eq})} = \cos(\eta_{|x-y|})\langle \hat{I}_{|x-y|}(0)\rangle_{\hat{\varrho}(0)} + i\sin(\eta_{|x-y|})\langle \hat{I}_{|x-y|}(\pi/2)\rangle_{\hat{\varrho}(0)} \,. \tag{20}$$

Here and throughout whenever $x, y$ are positions on the chain then $|x - y|$ is meant in the sense of the distance on the ring geometry. Note that due to the average the equilibrium covariance matrix and hence also $\hat{\varrho}_G^{(eq)}$ will be translation invariant which implies that the current operators can be evaluated by a strictly local measurement. For example if the initial covariance matrix is real then $\eta = 0$ and we have

$$I_{|x-y|} = \langle \hat{I}_{|x-y|} \rangle_{\hat{\varrho}_G^{(eq)}} = \langle \hat{f}_x^\dagger \hat{f}_y \rangle_{\hat{\varrho}_G^{(eq)}} + \text{h.c.} , \tag{21}$$

where $x, y$ can be chosen arbitrarily as long as their separation is $d = |x - y|$.

## 4 Power-law equilibration

### 4.1 Strategy of the argument

Our goal in this section is to bound how quickly time-evolved second moments $t \mapsto \Gamma_{x,y}(t)$ relax towards the time-averaged value. The culmination of this is a bound of the form

$$\left| \Gamma_{x,y}(t) - \Gamma_{x,y}^{(eq)} \right| \leq C_\Gamma t^{-\gamma} , \tag{22}$$

where $C_\Gamma, \gamma > 0$ are constants independent of the system size. Let us begin by defining the decomposition of the covariance matrix $\Gamma$ into its currents $\Gamma^{(d)}$ with entries

$$\Gamma_{x,y}^{(d)} = \Gamma_{x,y} \delta_{x,y+d} , \tag{23}$$

where we use the convention $\delta_{a,b+L} = \delta_{a,b}$. Intuitively, one can find $\Gamma^{(d)}$ by picking out bands from $\Gamma$ parallel to the diagonal and we will show that each band equilibrates individually to the conserved current value $I_d$ using that the evolution is linear in the bands $\Gamma(t) = \sum_{d=-\lfloor (L+1)/2 \rfloor + 1}^{\lfloor L/2 \rfloor} \Gamma^{(d)}(t)$. Now we expand $\Gamma^{(d)}$ via the discrete Fourier transform

$$\Gamma_{z+d,z} = \sum_{n=1}^{L} \mathcal{X}_n^{(d)} e^{2\pi i n z / L} . \tag{24}$$

Here $\mathcal{X}_n^{(d)}$ are defined implicitly by the inverse discrete Fourier transform and the most important one is

$$\mathcal{X}_{n=L}^{(d)} = \frac{1}{L} \sum_{x=1}^{L} \Gamma_{x,x+d} = \Gamma_{x,y}^{(eq)} , \tag{25}$$

which is the equilibrium value. After a technical calculation we obtain

$$\Gamma_{x,y}^{(d)}(t) = \sum_{n=1}^{L} \mathcal{X}_n^{(d)} e^{2\pi i n (x-d)/L} f_n(t) , \tag{26}$$

with

$$f_n(t) = \frac{1}{L} \sum_{k=1}^{L} e^{i(\omega_{(k+n)} - \omega_k)t + 2\pi i s(x-y-d)/L} . \tag{27}$$

This step is of crucial importance. We have separated out a dynamical function $f_n$ which, when it decays, does so independent of the initial state – or colloquially speaking, it scrambles the initial state. To prove our result, we show in Appendix C that $f_n$ dephases in time

$$|f_n(t)| \leq C_\# \left( \frac{n\pi}{L} \right) t^{-\gamma} , \tag{28}$$

with some constant $\gamma > 1/(6R+6)$. Here, one should note that $C_\#(n\pi/L)$ will be constant in time but could depend on the system size. Indeed for $n \approx 1$ we will have $C_\#(n\pi/L) \sim L^2$. However, we will see that this is not an artefact of the technique that we use to obtain the bound (28) – points $n$ with constant larger than some threshold $C_\#(n\pi/L) > C_{\text{th}}$ are *resilient points* where $f_n$ dephases slowly and will be discussed in detail below. In the end $C_\#$ has a simple form and it does not scale in the system size for very many natural initial configurations.

In order to derive the bound from Eq. (28), we will study the *phase function* $\Phi_{t,\alpha} : [0, 2\pi) \to \mathbb{R}$ defined as

$$\Phi_{t,\alpha}(p) = Dp - 4t \sum_{z=1}^{R} J_z \sin(z\alpha) \sin(zp + z\alpha) \, . \tag{29}$$

Choosing $D = 2\pi(x - y + d)/L$ and $\alpha = \pi n/L$ we have

$$f_n(t) = \frac{1}{L} \sum_{k=1}^{L} e^{i\Phi_{t,\alpha}(2\pi k/L)} \, . \tag{30}$$

This relation (30) is called an exponential sum and its dephasing is instrumental for the state to dephase itself. In order to bound it, we make use of the *Kusmin-Landau technique* [49]. This powerful machinery allows to arrive at quantitative bounds as opposed to intuitive estimates obtained from stationary phase approximations [80, 81]. The crux of this method is, however, similar – dephasing is determined by the *gaps* of $\omega$ or specifically by the first derivative of $\Phi$. By analyzing the dispersion relation $E$, we find a lower bound to the gaps by appropriate Taylor expansions. The bound is then determined by the values of the derivatives of $\Phi$ at points that one could view as stationary points. We define

$$\mathcal{S}_\alpha^{(1)} = \{p \in [0, 2\pi] \text{ s.t. } \Phi_{t,\alpha}'(p) = 0\} \, , \tag{31}$$

and correspondingly

$$\mathcal{S}_\alpha^{(2)} = \{p \in [0, 2\pi] \text{ s.t. } \Phi_{t,\alpha}''(p) = 0\} \tag{32}$$

for the second derivative. Due to the finite range $R$ of the Hamiltonian, there are at most $2R+2$ stationary points, which we prove in the appendix. While in the appendix we prove a more general statement, here we discuss the generic case only where we assume that there are no points such that $\Phi_{t,\alpha}''(p) = \Phi_{t,\alpha}'''(p) = 0$. Hence, for any first order root $r \in \mathcal{S}^{(1)}$ we either have $\Phi_{t,\alpha}''(r) \neq 0$ or $\Phi_{t,\alpha}'''(r) \neq 0$. For the Taylor expansion we want to take the value of the minimal derivative that does not vanish at $r$ so we take $\kappa_r = 1$ if $\Phi_{t,\alpha}''(r) \neq 0$ and otherwise we set $\kappa_r = 2$. In Appendix C we show a more general statement, but in the generic case we simply have

$$\gamma = 1/3 \tag{33}$$

and

$$C_\#(\alpha) = 6(2R + 1) \max\{1, 8R^4 J_{\text{max}}/M_\alpha^2\} \, , \tag{34}$$

where we define the minimal derivative value used for lower bounding dephasing through a Taylor expansion

$$M_\alpha = \frac{1}{t} \min \left\{ \min_{r \in \mathcal{S}^{(1)}} \left| \Phi_{t,\alpha}^{(\kappa_r+1)}(r) \right|, \min_{r \in \mathcal{S}^{(2)}} \left| \Phi_{t,\alpha}'''(r) \right|^2 \right\} \, . \tag{35}$$

Note that this constant is time independent hence the time scaling is governed by the smallest next order derivative which does not vanish at a stationary point.

Hence, as proved in Appendix C, we obtain a bound on the dephasing of the form (28), which is a huge simplification as the bound is now encoded in the minimal value of derivatives at stationary points which is a sparse set. As an example, let us study $M_\alpha$ of $\hat{H}(h)$ with only one non-trivial coupling value $J_1 \neq 0$. Then we have the simplification

$$\Phi'_{t,\alpha}(p) = D - 2tJ_1 \sin(\alpha)\cos(p+\alpha) . \tag{36}$$

Then we find that $\mathcal{S}^{(1)}$ has at most 2 roots and we should evaluate the value of the second derivative at these points

$$\Phi''_{t,\alpha}(p) = 2tJ_1 \sin(\alpha)\sin(p+\alpha) . \tag{37}$$

Now, we notice that for $n \approx 0$ we have $\alpha = n\pi/L \approx 0$ which means that $M_\alpha \sim \alpha \sim L^{-1}$ and hence $C_\# \sim L^2$ becomes size dependent. In this case $C_\#$ can be independent of the system size only if $n$ is a significant fraction of $L$. However, inspecting (27) for $\alpha = n\pi/L \approx 0$ we find that it will in fact not dephase for the same reason that our bound yields a large $C_\#(\alpha)$ constant as we have

$$f_n(t) \approx \frac{1}{L}\sum_{k=1}^{L} e^{2\pi i k(x-y+d)/L} \tag{38}$$

for times $t \ll L$. Therefore we would need times $t$ scaling in the system size for dephasing to even set in – this is an effect that we call *resilience*.

## 4.2 Definition of non-resilient second moments

Choosing the initial state such that $\Gamma$ has substantial $\mathcal{X}_n^{(d)}$ around a resilient point will render the covariance matrix resilient against equilibration. This should be expected and has been discussed in the literature [2] with the simplest example being a system with a linear dispersion relation. By Eq. (9) we see that generically we will find regions in momentum space where the dispersion relation is indeed approximately linear and populating the initial state with quasi-particles from these regions will obstruct dephasing. More generally, resilience to equilibration can be characterized within the framework of resource theories [82]. Here, we have enough structure to be able to phrase a sufficient condition for correlations to be non-resilient using the above intuition.

**Definition 2** (Non-resilient second moments). *For a threshold constant $C_{\text{th}} > 0$ independent of the system size $L$ we call points in*

$$\mathcal{R} = \{\alpha \in (0,\pi) \ s.t. \ C_\#(\alpha) \geq C_{\text{th}}\} \tag{39}$$

*resilient. If for all $d$ there exist constants $C_{RS}, C_{NRS} > 0$ independent of the system size such that the distribution $\mathcal{X}$ has little weight at resilient points*

$$\sum_{\frac{n\pi}{L} \in \mathcal{R}} \left|\mathcal{X}_n^{(d)}\right| \leq C_{RS} L^{-1} \tag{40}$$

*and is bounded outside*

$$\sum_{\frac{n\pi}{L} \notin \mathcal{R}} \left|\mathcal{X}_n^{(d)}\right| \leq C_{NRS} , \tag{41}$$

*then we say that the correlations $\Gamma$ are non-resilient second moments at the level $C_{\text{th}}$.*

The crucial mathematical feature of this definition that is needed to ensure equilibration is the system size independence of the constants such that constants derived in further bounds are also system size independent. Notice that in the definition of $\mathcal{R}$ we exclude $\alpha = \pi$ which corresponds to $\Gamma_{x,x+d}^{(eq)} = I_d = \mathcal{X}_{n=L}^{(d)}$ which is a constant of motion. In the following we will bound the deviation from equilibrium $|\Gamma_{x,y}(t) - \Gamma_{x,y}^{(eq)}|$ and hence this definition can be thought of as defining initial conditions that are non-resilient to equilibration towards *translation invariant* steady states.

## 4.3 Equilibration of non-resilient second moments

We can easily see that with this definition, we can give a bound as to how fast individual currents (26) relax as long using the bound (28) where now we have the promise that $C_\# \leq C_{th}$. Indeed, at the resilient points we can use a trivial upper bound $|f_n(t)| \leq 1$, to obtain

$$\left| \Gamma_{x,y}^{(d)}(t) - I_d \delta_{x,y+d} \right| \leq C_{RS} L^{-1} + C_{NRS} C_{th} t^{-\gamma} \tag{42}$$

$$\leq C_\Gamma^{(d)} t^{-\gamma}, \tag{43}$$

where in the second line we used $t \leq t_R = \Theta(L)$. By the decay of correlations only currents with range of the order of the correlation length $d_\xi(t) = \xi \ln(t^\gamma)$ will be relevant. In the appendix we show using the unitarity of the propagator that

$$\sum_{d=d_\xi(t)}^{\lfloor L/2 \rfloor} \left| \Gamma_{x,y}^{(d)}(t) \right| \leq \frac{C_{Clust}}{1 + e^{-1/\xi}} t^{-\gamma}, \tag{44}$$

and hence one easily arrives at a bound for fluctuations of the covariance matrix entries away from equilibrium

$$\left| \Gamma_{x,y}(t) - \Gamma_{x,y}^{(eq)} \right| \leq \sum_{d=-\lfloor (L+1)/2 \rfloor +1}^{\lfloor L/2 \rfloor} \left| \Gamma_{x,y}^{(d)}(t) - I_d \delta_{x,y+d} \right| \tag{45}$$

$$\leq C_\Gamma t^{-\tilde{\gamma}},$$

where $C_\Gamma$ is obtained by appropriately collecting the system size independent constants and $\tilde{\gamma} \approx \gamma$ is chosen such that $\ln(t^\gamma) t^{-\gamma} \leq t^{-\tilde{\gamma}}$ for all times of interest $t_0 \leq t \leq t_R$. The following proposition encapsulating these ideas is proven in full detail in Appendix E.

**Proposition 3** (Equilibration of second moments). *Consider a fermionic system with initially exponentially decaying correlations and non-resilient second moments $\Gamma$. Then there exist a constant relaxation time $t_0$ and a recurrence time $t_R = \Theta(L)$ such that, for all $t \in [t_0, t_R]$,*

$$\left| \Gamma_{x,y}(t) - \Gamma_{x,y}^{(eq)} \right| \leq C_\Gamma t^{-\gamma}, \tag{46}$$

*where $C_\Gamma, \gamma > 0$ are constants.*

As we will see, this general bound must have $\gamma \leq 1/2$ by giving a specific example with a tight relaxation scaling via the Bessel function asymptotics. On the other hand, we have that the exponent is lower bounded due to $\gamma \geq 1/(6R) - \varepsilon$ for any $\varepsilon > 0$, as explained in Appendix E.

### 4.4 Examples of non-resilient second moments

As the simplest example of non-resilient second moments, consider the covariance matrix $\Gamma^{(0,1)}$ of the charge-density wave corresponding to the Fock state vector $|0,1,0,1,\dots\rangle$ which will equilibrate under the nearest-neighbour model. More generally, if there is no shift symmetry of the dispersion relation any $P$-periodic configuration of currents will be non-resilient for intensive $P$ not scaling in the system size, see Appendix F. This continues to hold true even in the presence of sparse defect sites at random points. This is the most important case and captures the intuition about what physically one should expect to be necessary for equilibration, namely that the mass distribution (and concomitantly currents) are already distributed over the system, albeit with possibly intensive random configurations at microscopic scales.

On the other hand, a $P$-periodic state with extensive $P$ will be resilient and not relax towards a translation invariant steady state according to a power-law. Specifically, second moments of the form $\Gamma_{x,x} = 1$ for $x \le L/2$ and all other entries vanishing are resilient. Intuitively this is a block of particles over an extensive part of the system and is resilient because by the Lieb-Robinson bound one would need to wait to extensively long times for the current to become evenly distributed. Such a covariance matrix would violate our definition of non-resilient second moments already on the level of $\mathcal{X}_n^{(d)}$, see Appendix F. Let us finally remark that the definition of non-resilient second moments has a linear structure and mixtures of different $P$-periodic covariance matrices $\Gamma^{(P)}$ are again non-resilient, as long as the weights decay fast enough, i.e.,

$$\Gamma^{(\text{Mixt})} = \sum_P a_P \Gamma^{(P)} \tag{47}$$

can be non-resilient for various weights $a_P$.

If we would like to quantify the resilience in the generic case, we may neglect physical constraints on the covariance matrix and choose $\Gamma_{x,y}^{(\text{rnd})} \in [a,b]$ uniform at random. In this case, we will indeed find non-resilience on average $\mathbb{E}[\mathcal{X}_n^{(d)}] = (a+b)\delta_{n,L}/2$. However, the fluctuations are rather large as we find $\text{Var}[\mathcal{X}_n^{(d)}] = (a-b)^2/(12L)$, so drawing a random selection from the uniform distribution will often yield a significant number of the $L$-many harmonics to be of the order $(X_n^{(d)})^2 \sim \text{Var}[\mathcal{X}_n^{(\text{rnd},d)}] \sim L^{-1}$ which is too large and could lead to resilience. Yet, constructing a mixture of such matrices can smoothen the distribution and so for $\Gamma = \sum_{k=1}^{K} \Gamma^{(\text{rnd}:k)}/K$, we should find $\mathcal{X}_n^{(d)} \approx \mathbb{E}[\mathcal{X}_n^{(d)}]$ up to fluctuations decaying $K^{-1/2}$, i.e., one can get closer to the average behaviour which is non-resilient. Observe, that by Eq. (27) dephasing could also occur if the Fourier weights $\mathcal{X}_n^{(d)}$ are larger than what we allow for in Definition 2 if they fluctuate uniformly on the scale where $f_n(t)$ does not change strongly. Later, in order to discuss equilibration of a random selection of second moments, which are physically admissible and have a finite correlation length, we will discuss thermal states of the Anderson insulator – numerically we indeed find equilibration in that case too.

Finally, note that our definition of non-resilient second moments characterizes initial states that equilibrate to translation invariant steady states. However, it is important to note that non-translation invariant steady states can also occur – due to possible shift symmetries of the dispersion relation such that we have $\omega_k = \omega_{k+n}$ for all $k = 1,\dots,L$. The simplest example is to notice that $\omega_k = \omega_{k+L/2}$ for the next-nearest-neighbour model so then $f_{L/2}(t) = \texttt{const}$. In this case, our definition of non-resilient second moments excludes any $\Gamma$ which has significant $\mathcal{X}_n^{(d)}$ for $n \approx L/2$ via the condition on the $C_\#(n\pi/L) \le C_{\text{th}}$ constant. These are very special cases, see Fig. 2 and we have chosen to study equilibration exclusively towards translation invariant steady states. Notably, the nearest-neighbour model has no shift symmetry hence only states with long-range dislocations, or a population of long-wavelength quasiparticles are being excluded by the definition of non-resilience.

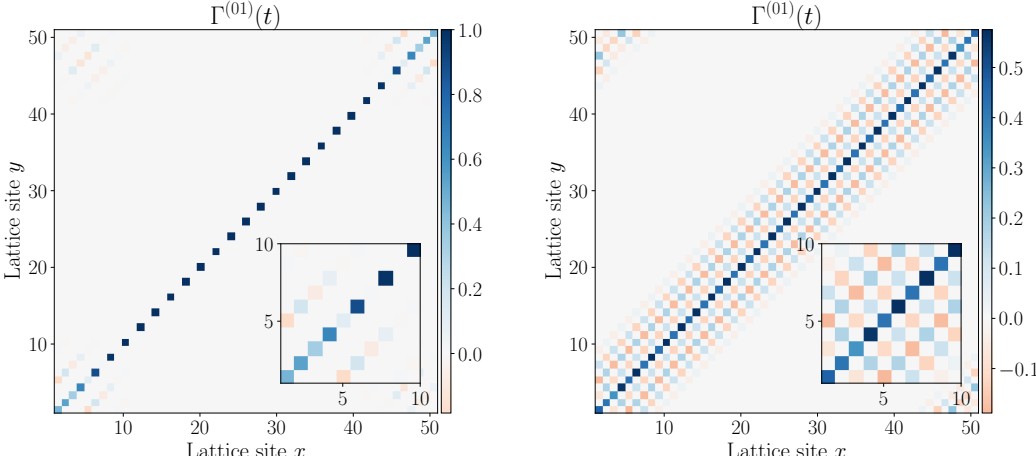

Figure 2: Covariance matrix of a charge-density wave $\Gamma^{(0,1)}$ which corresponds to the Fock state vector $|0, 1, 0, 1, \dots\rangle$ has varying equilibration behaviour depending on the locality of the Hamiltonian and the system size parity. For the next-nearest-neighbour model the system in this special initial state splits up into two independent sub-lattices and is in an exact steady state whenever the system size $L$ is even. However, for odd $L$ the symmetry of the density distribution is incommensurate with the system size and there is necessarily a defect of the type $|\dots, 0, 1, 0, 0, 1, \dots\rangle$ or $|\dots, 1, 0, 1, 1, 0, 1, \dots\rangle$ around which the charge-density wave pattern starts becoming homogeneous. Note, that away from the defect point the charge-density wave looks locally like a steady state of the Hamiltonian and one can prove by the LR bounds that the middle region will remain unaffected for extensively long times. The left plot shows $\Gamma^{(0,1)}(t = 1.5)$ after a quench to the next-nearest-neighbour model (the inset throughout shows the sub-block of the first 10 sites). On the other hand, if we quench to the nearest-neighbour model then there is no transient symmetry present and the charge-density wave is completely non-resilient and homogeneously tends towards equilibrium as seen in the right plot for the same initial state.

## 4.5  $P$-periodic initial density distributions and nearest neighbour hopping

A specifically instructive case is to study the situation in which the initial state is such that the covariance matrix is diagonal with a $P$-periodic structure, and the system is quenched to evolve via the nearest neighbour hopping model. This means that the density distribution repeats every $P$ sites in that $\Gamma_{x,x} = \Gamma_{x+P,x+P}$ for all $x$. It is one of the strengths of our result that we need not care about the structure within the block because any such distribution for an intensive $P$ is non-resilient. The steady state will be translation invariant and diagonal with the second moments given by $\Gamma^{(eq)}_{x,y} = \delta_{x,y}/F$ where $1/F$ is the filling ratio. For example, if the initial covariance matrix was $\Gamma^{(1,0)} = \mathrm{diag}(1, 0, 1, 0, \dots)$, then we have half-filling $1/F = 1/2$ and for $\Gamma^{(1,0,0)} = \mathrm{diag}(1, 0, 0, 1, 0, 0\dots)$ we get $1/F = 1/3$. When considering the evolution under a nearest-neighbour fermionic hopping Hamiltonian, and such initial conditions, we find that the propagator follows a law $O(t^{-1/2})$ in time, as laid out in Appendix A.3, dictated by the asymptotics of Bessel functions of the first kind. This decay is inherited by the actual correlation decay, in that for any $P$-periodic initial condition, one finds that

$$|\Gamma_{x,y}(t) - \Gamma^{(eq)}_{x,y}| = O(t^{-1/2}) \tag{48}$$

for all $x, y$. It is also interesting to note that the resulting steady states can be seen as an *infinite temperature Gibbs state* at a specific chemical potential which imposes the value of the

total particle number. Specifically, one finds for the equilibrium covariance matrices

$$\Gamma_{x,y}^{(eq)} = \frac{1}{1 + e^{-\mu}} \delta_{x,y} \,, \tag{49}$$

from which one can obtain the value of the chemical potential in explicit form $\mu = -\ln(F-1)$ for any $x, y$ due to translation invariance.

# 5 Quasi-free ergodicity

## 5.1 Notions of ergodicity

One of the key questions of statistical mechanics is what precise properties of the Hamiltonian governing the dynamics can be held responsible for the emergence of aspects of quantum statistical mechanics. In classical mechanics it results from sufficient transport properties which is evident already in Boltzmann's H-Theorem. In the quantum regime for free systems, a notion with similar operational meaning can also be identified, namely that propagators decay quickly, which holds with surprising generality and can be interpreted as a lower bound to particle transport.

**Theorem 4** (Free fermionic ergodicity). *Let $t \mapsto G(t)$ be the propagator for a non-interacting translation invariant fermionic Hamiltonian $\hat{H}(h)$ which is off-diagonal on the one-dimensional real-space lattice. Then for all times $t$ between a relaxation time $t_0 = O(1)$ up until a recurrence time $t_R = \Theta(L)$ the propagator obeys*

$$|G_{x,y}(t)| \leq C t^{-\gamma} \,, \tag{50}$$

*where $C, \gamma > 0$ are constants. We can take $\gamma = 1/3$, provided there are no points $p$ such that $E''(p) = E'''(p) = 0$ which is true for generic models.*

We can interpret Theorem 4 as proving free-particle ergodicity for these models. This notion of ergodicity is motivated by the classical notion of ergodicity, which states that an ergodic system essentially explores the whole available phase space, and it does so homogeneously. In free systems, we have to respect the linear constraint in the relation (10) at all times and given that, the suppression (50) allows to show that the particles must spread over the lattice. Indeed unitarity of the propagator $\sum_{y=1}^{L} |G_{x,y}(t)|^2 = 1$ implies that a particle initially at site $x$ must occupy at least $O(t^{2\gamma})$ sites. If for most sites the bound is not tight, then the particle must have spread to an even larger region. Indeed, whenever the spatial separation $d = x - y$ is far away from a ballistic wavefront, typically found in free translation invariant systems, then our proof can be used to obtain $\gamma = 1/2$ which would imply that the particle spreads homogeneously over a region of size $O(t)$. Note that our bound is independent of $d$ and hence $\gamma = 1/3$ is necessary and reflects the scaling at the wavefront [50]. Conversely, in localized systems such as Anderson insulators, particles cannot spread freely and typically $|G_{x,y}(t)| \leq C e^{-|x-y|/\ell_0}$ which together with the unitarity of the propagator can be used to show that $\sum_{x=y-\ell_0}^{y+\ell_0} |G_{x,y}(t)| > O(1)$ for all times, i.e., particles cannot spread by more than the localization length $\ell_0$. The proof of Theorem 4 is given in Appendix D and is again based on Kusmin-Landau inequality [49]. It could be also of general interest as a method for deriving error-bars for stationary phase arguments in field theory. Note that one can explicitly calculate the relaxation time and the recurrence time, $t_0$ and $t_R$, using only the dispersion relation.

## 5.2 Gaussification is generic

Combining the above results with insights from Ref. [50] lead to a remarkably strong result. Ref. [50] presented results on how non-interacting fermionic quantum systems that show delocalizing transport would "Gaussify", that is, turn to a quantum state that is Gaussian to an arbitrarily good approximation in time. However, Theorem 4 shows precisely this: Non-interacting one-dimensional models generically exhibit delocalizing transport. We hence arrive at a statement of a rigorous convergence to a generalized Gibbs ensemble with enormous generality. When stating this Gaussification theorem, we define the state $\hat{\varrho}_G(t)$ to be a Gaussian state with the same covariance matrix as $\hat{\varrho}(t)$.

**Theorem 5** (Fermionic generic Gaussification)**.** *Consider the initial fermionic state $\hat{\varrho}(0)$ with exponential decay of correlations and a non-interacting translation-invariant post-quench Hamiltonian with dispersion relation $E(p)$ such that there are no points with $E''(p) = E'''(p) = 0$ for any $p$. Then there exist a constant relaxation time $t_0$ and a recurrence time $t_R = \Theta(L)$ such that, for all $t \in [t_0, t_R]$,*

$$|\langle \hat{A} \rangle_{\hat{\varrho}(t)} - \langle \hat{A} \rangle_{\hat{\varrho}_G(t)}| \leq C t^{-1/6} , \tag{51}$$

*where $C > 0$.*

# 6  Proving Theorem 1

In this section we collect all our findings that lead to the statement of Theorem 1. Within the setting described above the two crucial ingredients are an initial state featuring exponentially decaying correlations and the quench Hamiltonian being translation invariant. By Theorem 5, we have that at a sufficiently large time any local correlation function can be approximated by the value obtained from the Gaussified state. That is, it suffices to take the second moments $\Gamma$ of the initial state $\hat{\varrho}(0)$, evolve them according to the quench Hamiltonian and evaluate $\langle \hat{A} \rangle_{\hat{\varrho}(t)}$ by appropriately employing Wick's theorem for $\Gamma(t)$. We hence find equilibration $\langle \hat{A} \rangle_{\hat{\varrho}(t)} \approx \mathtt{const}$ if $\Gamma(t) \approx \mathtt{const}$ is time independent. This is already the case if we perform the quench starting from a translation-invariant non-Gaussian state because then the covariance matrix $\Gamma$ is also translation invariant and so $\Gamma(t) = \Gamma(0)$ because $\partial_t \Gamma(t) = i[h, \Gamma(t)] = 0$. For such cases Gaussification is sufficient for equilibration [50]. However, thanks to Proposition 3 we obtain a much more general statement. Namely, any non-resilient covariance matrix will equilibrate. This result applies to very natural initial conditions that can dramatically deviate from a homogenous configuration. The relaxation takes the form of a power law $O(t^{-1/6})$ determined by the Gaussification times. This, however, is an artifact of our rigorous uniform bounds – one should expect the calculation for a special configuration from Refs. [42, 54] to be generic $O(t^{-1/2})$. A proof of such a behaviour being the standard time-scale may be possible but would involve a significantly more detailed treatment of the wavefront which is responsible for our scalings not being tight as compared to the behaviour in the bulk of the Lieb-Robinson cone [50].

# 7  Numerical results

## 7.1 Quenches of the Anderson insulator to an ergodic translation invariant Hamiltonian

As a numerical illustration, in this section, we discuss the situation arising from starting in the thermal state of a disordered Anderson insulator, initially not translation invariant, followed

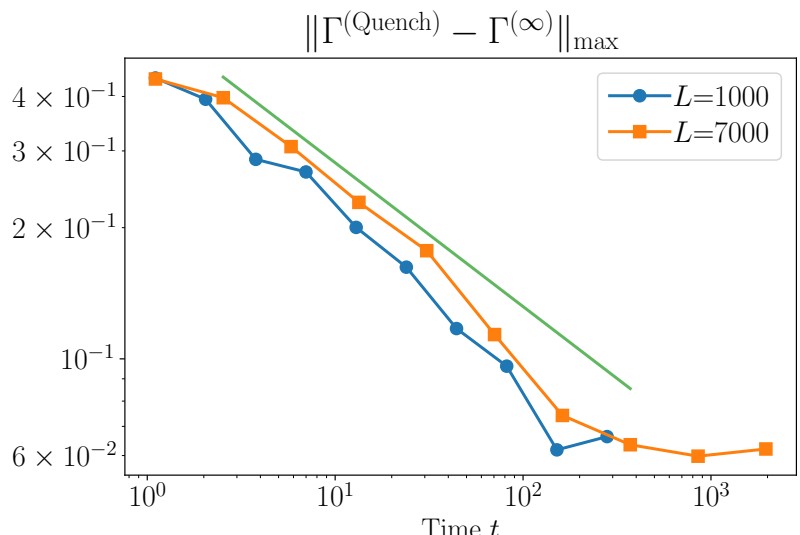

Figure 3: We have sampled a thermal state of the Anderson insulator $\Gamma^{(\text{Quench})}$ for system sizes $L = 1000, 7000$ at $\beta = 1$ and $w = 5$ as an example of a strongly disordered initial condition. We find that after switching off the on-site disorder the ensuing non-equilibrium evolution under the nearest-neighbour hopping model leads to relaxation towards the infinite-time average $\Gamma^{(\infty)}$ which is indeed quantified in the functional form by a power-law in time $\|\Gamma^{(\text{Quench})} - \Gamma^{(\infty)}\|_{\max} \sim t^{-\alpha}$. The green line is a guide to the eye scaling as $\sim t^{-1/3}$. At some point, the power-law relaxation must level off either due to finite system size, with the ultimate small parameter being $\epsilon \sim t_{\text{R}}^{-\alpha} \sim L^{-\alpha}$, or due to the specific quasiparticle content $\mathcal{X}^{(d)}$.

by a quench to a perfectly translation invariant ergodic hopping Hamiltonian. Needless to say, the equilibrium states emerging are once again generalized Gibbs ensembles and Gaussian states: It is interesting to note, however, that they resemble fully thermal states with high probability to a rather good approximation.

To be specific, as a starting point we choose an initial covariance matrix which is not translation invariant and has a finite correlation length. A natural way of assigning such initial conditions is to consider a Gibbs state of the Anderson insulator with Hamiltonian

$$\hat{H}_\xi = \sum_{x=1}^{L} \left( \hat{f}_{x+1}^\dagger \hat{f}_x + \hat{f}_x^\dagger \hat{f}_{x+1} + \xi_x \hat{f}_x^\dagger \hat{f}_x \right), \tag{52}$$

where the noise is uniformly distributed in the interval $\xi_x \in [-w, w]$ for $w > 0$. We study the quench consisting in switching off the disorder, i.e., setting $\xi_x = 0$ for all $x$. Following a numerical calculation, the quenched state $\Gamma^{(\text{Quench})}(t) = \Gamma^{(\beta, \text{Anderson})}(t)$ can be seen to become largely homogeneous for sufficiently long duration of the evolution, see Fig 3. As a measure of equilibration, we make use of the max norm distance

$$\|\Gamma^{(1)} - \Gamma^{(2)}\|_{\max} = \max_{x,y} \left| \Gamma_{x,y}^{(1)} - \Gamma_{x,y}^{(2)} \right| \tag{53}$$

between two covariance matrices $\Gamma^{(1)}, \Gamma^{(2)}$. Whenever $\|\Gamma^{(1)} - \Gamma^{(2)}\|_{\max}$ is small, a large fidelity between the two states is implied [83, 84]. Fig. 3 provides further substance to the above established rigorous insights, in that a significant part of the equilibration is indeed governed by a power-law by comparing $\Gamma^{(\text{Quench})}(t)$ to the infinite time average $\Gamma^{(\infty)}$. To further elaborate on this setting, we discuss the features of the equilibrium state, see Fig. 4. We begin by

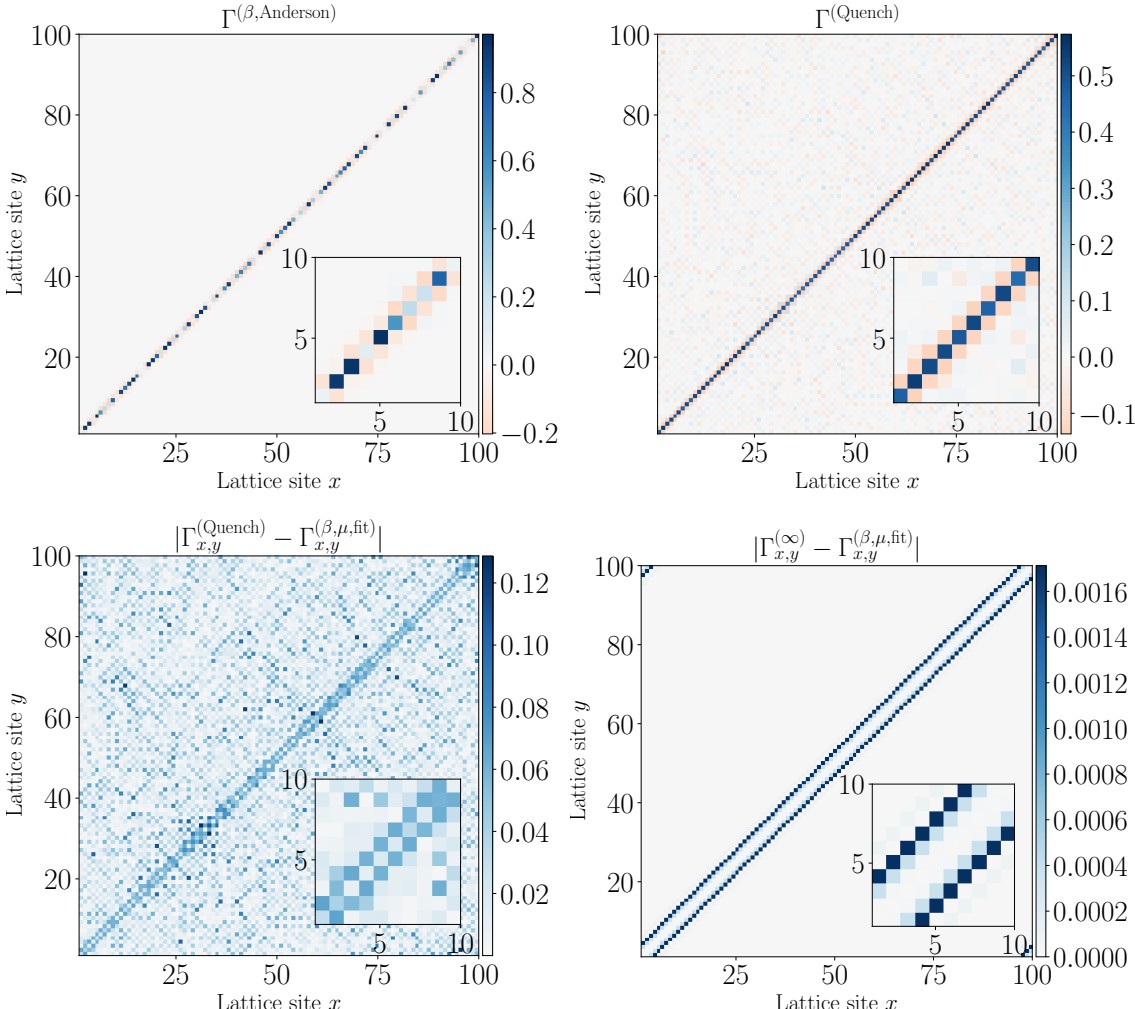

Figure 4: A system initially in a thermal state of an Anderson insulator $\Gamma^{(\beta,\text{Anderson})}$ with $\beta = 1$ and $w = 5$ (top-left) can be quenched to translation invariant evolution by switching off the on-site disorder which results in approximate translation invariance $\Gamma^{(\text{Quench})} = \Gamma^{(\beta,\text{Anderson})}(t = L/4)$ (top-right). This can be quantified with a comparison to the thermal state of nearest-neighbour hopping $\Gamma^{(\beta,\mu,\text{fit})}$ obtained from fitting over the temperature $\beta$ and chemical potential $\mu$ (bottom-left). While deviations are seen the inverse system size $L^{-1} = 10^{-2}$ is not a stringent small parameter. However if equilibration occurs for larger systems, then it will be towards the infinite time average $\Gamma^{(\infty)}$ which looks thermal at already small system sizes (bottom-right) and this property is retained when going towards the thermodynamic limit.

investigating the difference between the quenched state $\Gamma^{(\text{Quench})}(t)$ and a fit to a thermal covariance matrix $\Gamma^{(\beta,\mu,\text{fit})}$ of the quench Hamiltonian obtained from fitting over the temperature $\beta$ and chemical potential $\mu$. We find that the discrepancy is already diminished for $L = 100$ and $|\Gamma^{(\text{Quench})}_{x,y}(t) - \Gamma^{(\infty)}_{x,y}|$ is homogeneously distributed. For the infinite time average we have $\|\Gamma^{(\infty)} - \Gamma^{(\beta,\mu,\text{fit})}\|_{\max} \approx 10^{-3}$ as the distance to the Gibbs state. The upshot of the findings is that due to a concentration of measure effect, the resulting generalized Gibbs ensembles are with high probability close to an actual Gibbs state, with stray fluctuations being detected. We discuss further details of this argument in Appendix F.5.

## 7.2 Realizing generalized Gibbs ensembles in optical lattices

Ultra-cold atoms trapped in optical lattices [85] have proven to be an excellent platform for studying relaxation phenomena [86, 87] in instances of quantum simulators [88–90], because the system is well isolated from the environment during the evolution and one can prepare with high-level of control states that have very visible non-equilibrium dynamics after the quench. Here we hint that with the present techniques that have been used in various settings one can prepare two different initial states that will equilibrate to two different steady states which are easily distinguishable – despite the Hamiltonian governing the dynamics being the same in both cases. This is expected to be possible at least for intermediate times in instances of *prethermalization*, before interaction effects will lead to a genuine full thermalization. The first steady state would be one obtained from simply letting the gas equilibrate on the lattice. For the second type of the steady state, we would prepare the initial state in the same way and perform the quench by suddenly doubling the lattice by adding in-between sites, exploiting optical *super-lattices*, similar to the situation described in Ref. [86]. In that situation the initial covariance matrix will feature a checker-board pattern with only the odd sites being occupied and currents being non-zero only between odd sites, see Fig. 5. Note that the specific details of how the doubling is performed are not important as long as the initial state preparation will feature a charge-density wave pattern – however it is absolutely crucial for our example that the charge-density wave is also present in the current structure. The quench then consists in allowing for tunneling between all sites. By our analytical result, the density pattern which is a $P = 2$-periodic block structure will equilibrate to a uniform distribution at each site. The same, again, will occur for each current individually. Usually, the nearest-site tunnelling current will be the strongest so if we had $I_1 = \langle \hat{I}_1 \rangle$ before the doubling then the current will equilibrate to $I_1/2$ after the doubling of the lattice. However, the surprise value lies in the fact that this will be the next-nearest-neighbour current $I_2'$ in the new lattice, and in the steady state the final nearest-neighbour current should not be present $I_1' \approx 0$. That is, after the quench, one will observe that there are only currents in the system in multiples of two sites, cf. Fig. 5. This is a non-trivial observation, because the sites that have been un-occupied immediately after the doubling will become occupied and there will be currents flowing out of them to the next-nearest sites, i.e., the neighbouring initially un-occupied sites. In contrast, in the steady state there will be no tunnelling between the nearest-neighbour sites which is unintuitive as the Hamiltonian is nearest-neighbour showcasing peculiar memory effects that can be obtained with quenches to quasi-free evolution. Realizing such a setup in gases where interactions can be controlled by a Feshbach resonance would also allow to study how nearest-neighbour currents can be generated by many-body scattering, an effect not present in a non-interacting Hamiltonian [91–94].

## 8 Discussion and outlook

In this work, we have established a widely applicable and very general situation in which the convergence to generalized Gibbs ensembes can be proven. Specifically, we have shown in large generality that for large classes of natural initial conditions, local expectation values of systems relaxing under unitary dynamics generated by non-interacting Hamiltonians take the values of translation invariant genaralized Gibbs ensembles. The emerging steady state is parametrized by thermodynamical potentials whose number is intensive, namely of the order of the initial correlation length in units of the lattice spacing. Our assumption is that the quadratic Hamiltonian is translation invariant which leads to homogeneous spreading of particles on the lattice, a generic effect which we describe as a possible notion of ergodicity for quasi-free quantum systems. We have given numerical examples illustrating our rigorous

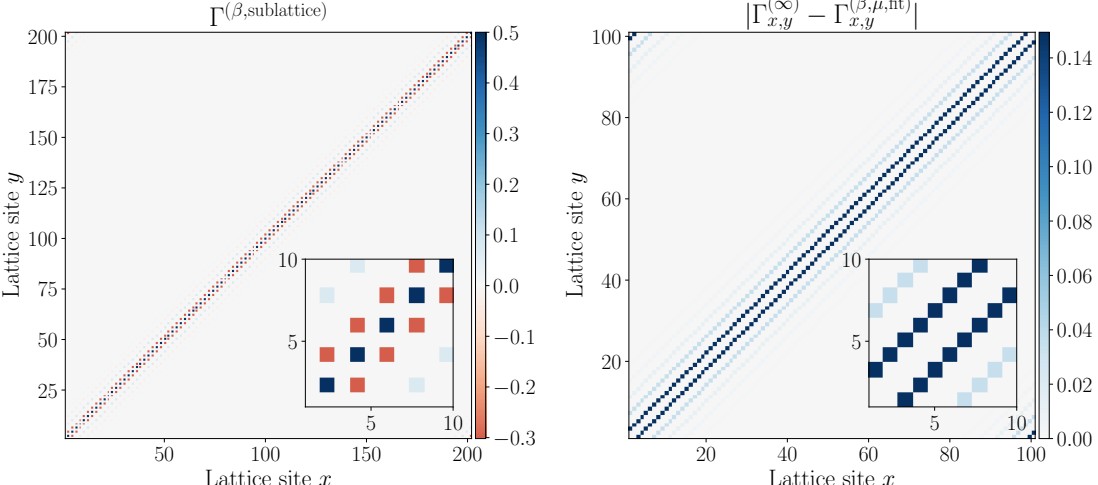

Figure 5: A system initially in a thermal state of the nearest-neighbour hopping Hamiltonian (left) on a sub-lattice can be quenched to translation invariant evolution which results in approximate translation invariance as one relaxes towards the steady state in finite time. The special initial condition results in the absence of nearest-neighbour currents on the whole lattice in the infinite time average (right). The best fit to a thermal state is given by an infinite temperature state with the corresponding filling ratio and strongly deviates from the steady state $\Gamma^{(\infty)}$. This is despite the density distribution becoming homogenous because the state deviates from the thermal ensemble by the absence of the nearest-neighbour current $I_1'$ and the presence of the next-nearest neighbour tunnelling $I_2'$. Note that there are particles and currents present on the initially unoccupied sub-lattice too. This showcases a general approach to creating initial conditions that demand a description in terms of a GGE by exploiting the existence of memory in terms of conserved local currents.

statements and explain how to observe non-trivial generalized Gibbs ensembles in, e.g., an optical lattice experiment.

Specifically, we saw that locally the memory of initial, possibly non-Gaussian, correlations is lost via the process of Gaussification, which relies only on finite correlation length in the initial state. Hence, even if the initial state preparation involves intricate interactions, a quench to quasi-free evolution will lead to a loss of memory of these initial strong correlations, and the state will obey Wick's theorem up to an error decaying algebraically in time. Such states (i.e., Gaussian) are determined only by their covariance matrix which we show to equilibrate. A necessary condition for this was that the initial current and density distributions did not have large-scale structure (which may still equilibrate, but only after a time of the order of the system size [38]). Thus, we derived a rapid polynomial time-scale for equilibration (which is independent of the system size). More precisely, the deviation from equilibrium of any normalized local correlation function is bounded by $\epsilon = O(t^{-\gamma})$, and the scaling is functionally tight, which we showed numerically.

The goal of our work was to show that it is possible to make rigorous statements concerning the dynamical emergence of statistical mechanics in mean-field models. For this reason we had to leave several aspects of the subject unanswered. Within our setting we have not discussed in detail the possibility of the infinite-time dephasing leading to steady states which are non-translation invariance due to degeneracy of the dispersion relation, and the proof of Lemma 11 in the appendix hints at that. It would also be interesting to understand in more detail if

Gaussification is possible for Green's functions which have only very weak quasi-free ergodicity, i.e., $|G_{x,y}(t)| = O(t^{-\gamma})$ for $\gamma < 1/4$ for a significant number of entries $x, y$. If the argument in Ref. [50] is optimal then one should observe for $\gamma < 1/4$ a temporal persistence of deviations from Wick's theorem for quenches of non-Gaussian states.

Concerning the question of adding small interactions one would expect the GGE examples that we have given to eventually thermalize. Understanding the dynamical stability of the GGE description is important for applications, e.g., work extraction protocols [95] but also is instrumental for our conceptual understanding of the emergence of thermalization. Above, we have hinted at an open problem of characterizing the structure of dephased states as being thermal in light of computational complexity and that an interesting approach would be to first make progress concerning high-temperature quenches.

## Note added

Upon completion of this manuscript, a preprint presenting closely related results appeared [96]. Our work puts significantly more emphasis on including rigorous error bounds, whereas Ref. [96] stresses more the physical intuition underlying the phenomena observed. The methods are also somewhat different (though related in spirit), as Ref. [96] uses stationary phase approximations, while we employ the machinery of Kusmin-Landau bounds.

## Acknowledgements

We gratefully acknowledge insightful discussions with M. Friesdorf, C. Krumnow, C. Gogolin, M. Goihl and T. J. Osborne.

**Funding information**   We thank the ERC (TAQ), the DFG (CRC 183, EI 519/14-1, EI 519/7-1, FOR 2724), and the Templeton Foundation for support. This work has also received funding from the European Union's Horizon 2020 research and innovation programme under grant agreement No 817482 (PASQUANS).

## A   Quasi-free propagators generated by non-interacting Hamiltonians

### A.1   Bosonic and fermionic lattice models

In this section we will derive the propagator representation from the main text. All statements concern quasi-free Hamiltonians conserving the total particle number.

**Fermions.**   A fermionic annihilation operator acting on mode $x$ is denoted by $\hat{f}_x$. These operators obey the canonical anti-commutation relations $\{\hat{f}_x, \hat{f}_y^\dagger\} = \hat{f}_x \hat{f}_y^\dagger + \hat{f}_y^\dagger \hat{f}_x = \delta_{x,y}$ and $\{\hat{f}_x, \hat{f}_y\} = \{\hat{f}_x^\dagger, \hat{f}_y^\dagger\} = 0\}$. Quasifree fermionic Hamiltonians conserving the particle number are of the form

$$\hat{H}(h) = \sum_{x,y}^{L} h_{x,y} \hat{f}_x^\dagger \hat{f}_y \,, \tag{54}$$

where $h = h^\dagger \in \mathbb{C}^{L \times L}$ is the coupling matrix for a finite system size $L$.

**Lemma 6** (Fermionic propagator). *We have*

$$\hat{f}_x(t) = e^{it\hat{H}(h)}\hat{f}_x e^{-it\hat{H}(h)} = \sum_{y=1}^{L} G^*_{x,y}(t)\hat{f}_y \,, \tag{55}$$

*where propagator is given by* $G^*(t) = e^{-ith}$.

*Proof.* We begin by noticing that $\hat{f}_x(t)$ is differentiable and take a time-derivative obtaining

$$\partial_t \hat{f}_x(t) = i\hat{H}(h)\hat{f}_x(t) - i\hat{f}_x(t)\hat{H}(h) \tag{56}$$

$$= i[\hat{H}(h), \hat{f}_x(t)], \tag{57}$$

which is the Heisenberg equation of motion. We further notice that

$$\partial_t \hat{f}_x(t) = i\, e^{it\hat{H}(h)} \, [\hat{H}(h),\, \hat{f}_x\,] \, e^{-it\hat{H}(h)} \,, \tag{58}$$

which means that we need to evaluate the commutator at $t = 0$. Next, we calculate the commutator

$$[\,\hat{f}_y^\dagger \hat{f}_z,\, \hat{f}_x\,] = \hat{f}_y^\dagger[\,\hat{f}_z,\, \hat{f}_x\,] + [\,\hat{f}_y^\dagger,\, \hat{f}_x\,]\hat{f}_z \tag{59}$$

$$= 2\hat{f}_y^\dagger \hat{f}_z \hat{f}_x + 2\hat{f}_y^\dagger \hat{f}_x \hat{f}_z - \delta_{x,y}\hat{f}_z \tag{60}$$

$$= -\delta_{x,y}\hat{f}_z \,, \tag{61}$$

which gives by linearity

$$[\hat{H}(h),\, \hat{f}_x] = -\sum_{y,z=1}^{L} h_{y,z}\delta_{x,y}\hat{f}_z \tag{62}$$

$$= -\sum_{z=1}^{L} h_{x,z}\hat{f}_z \,. \tag{63}$$

This allows us to write the above Heisenberg equation of motion explicitly as

$$\partial_t \hat{f}_x(t) = -i\sum_{y=1}^{L} h_{x,y}\hat{f}_y \,. \tag{64}$$

This is a system of $L$ linearly coupled ordinary differential equations and is solved by

$$\hat{f}_x(t) = \sum_{y=1}^{L} G^*_{x,y}(t)\hat{f}_y \,, \tag{65}$$

where $G^*(t) = e^{-ith} \in U(L)$. Indeed, this becomes apparent if one considers a vector $\hat{f} = (\hat{f}_1, \ldots, \hat{f}_L)^\top$ then we get in vector notation

$$\partial_t \, \hat{f}(t) = -ih \, \hat{f}(t) \quad \Longleftrightarrow \quad \hat{f}(t) = e^{-ith} \, \hat{f} = G^*(t) \, \hat{f} \,. \tag{66}$$

$\square$

**Bosons.** Bosonic operators $\hat{b}$ obey the canonical commutation relations $[\hat{b}_x, \hat{b}_y^\dagger] = \hat{b}_x \hat{b}_y^\dagger - \hat{b}_y^\dagger \hat{b}_x = \delta_{x,y}$ and $[\hat{b}_x, \hat{b}_y] = [\hat{b}_x^\dagger, \hat{b}_y^\dagger] = 0$. Quasifree bosonic Hamiltonians conserving the particle number are of the form

$$\hat{H}(h) = \sum_{x,y}^{L} h_{x,y} \hat{b}_x^\dagger \hat{b}_y \,, \tag{67}$$

where $h = h^\dagger \in \mathbb{C}^{L \times L}$ is again the coupling matrix for a finite system size $L$.

**Lemma 7** (Bosonic propagator)**.** *We have*

$$\hat{b}_x(t) = e^{it\hat{H}(h)} \hat{b}_x e^{-it\hat{H}(h)} = \sum_{y=1}^{L} G_{x,y}^*(t) \hat{b}_y \,, \tag{68}$$

*where the propagator is given by* $G^*(t) = e^{-ith}$.

*Proof.* Again, the Heisenberg equation of motion is

$$\partial_t \hat{b}_x(t) = i[\hat{H}(h), \, \hat{b}_x(t)] \tag{69}$$

and it suffices to evaluate the commutator at $t = 0$. We have

$$[\hat{b}_y^\dagger \hat{b}_z, \, \hat{b}_x] = [\hat{b}_y^\dagger, \, \hat{b}_x] \hat{b}_z \tag{70}$$

$$= -\delta_{x,y} \hat{b}_z \,, \tag{71}$$

which gives by linearity

$$\partial_t \hat{b}_x(t) = -i \sum_{y=1}^{L} h_{x,y} \hat{b}_y \,. \tag{72}$$

This is again a system of $L$ linearly coupled ordinary differential equations with the solution

$$\hat{b}_x(t) = \sum_{y=1}^{L} G_{x,y}^*(t) \hat{b}_y \,, \tag{73}$$

where $G = e^{-ith} \in U(L)$. Here, we have used the general correspondence

$$\partial_t \, \hat{b}(t) = -ih \, \hat{b}(t) \quad \Longleftrightarrow \quad \hat{b}(t) = e^{-ith} \, \hat{b} \tag{74}$$

for the vector $\hat{b} = (\hat{b}_1, \ldots, \hat{b}_L)^\top$.

$\square$

**Translation invariance.** Let us consider

$$\hat{H}(h) = \sum_{x,y}^{L} h_{x,y} \hat{a}_x^\dagger \hat{a}_y \,, \tag{75}$$

where $\hat{a}$ stands either for $\hat{f}$ in the case of fermions or $\hat{b}$ for bosons and $h = h^\dagger \in \mathbb{C}^{L \times L}$ is the coupling matrix for a finite system size $L$. The above two paragraphs have shown that

$$\hat{a}_x(t) = e^{it\hat{H}(h)} \hat{a}_x e^{-it\hat{H}(h)} = \sum_{y=1}^{L} G_{x,y}^*(t) \hat{a}_y \,. \tag{76}$$

In this paragraph we will be interested in translation invariant Hamiltonians.

**Lemma 8** (Translation invariant propagator). *Let $h$ be real translation invariant couplings with hopping amplitudes $J_k$. The propagator is given by*

$$G^*_{x,y}(t) = \frac{1}{L} \sum_{k=1}^{L} e^{-i\omega_k t + 2\pi i k(x-y)/L} \, , \tag{77}$$

*where $\omega_k = J_0 + 2 \sum_{z=1}^{\lfloor L/2 \rfloor} J_z \cos(2\pi k z/L)$.*

*Proof.* A translation invariant model has couplings which satisfy $h_{x,y} = h_{x+z,y+z}$ with periodic boundary conditions. Below we recall that such matrices are called circulant and are diagonalized by a discrete Fourier transform. Hence we can write

$$h_{x,y} = \frac{1}{L} \sum_{k=1}^{L} \omega_k e^{2\pi i k(x-y)/L} \, , \tag{78}$$

with $\omega_k$ as above which is obtained by an explicit calculation using Fourier modes. Using the formula $G(t) = e^{-ith}$, we hence get

$$G^*_{x,y}(t) = \frac{1}{L} \sum_{k=1}^{L} e^{-i\omega_k t + 2\pi i k(x-y)/L} \tag{79}$$

for the propagator. $\qquad\square$

## A.2 Circulant matrices

In this section we gather some basic facts about circulant matrices, leading up to the characterization that these are exactly the matrices diagonalizable by a discrete Fourier transformation. Additionally we describe simple formulas for the spectrum in the general case and for periodic boundary conditions. We begin by giving a precise definition of a circulant matrix.

**Definition 9** (Circulant matrix). *A matrix $h \in \mathbb{C}^{L \times L}$ is called circulant if*

$$h_{x,y} = h_{x+z,y+z} \tag{80}$$

*for any $x, y, z = 1, \dots, L$ [1] and we use modulo-L indices i.e. $h_{x+L,\cdot} = h_{x,\cdot}$ and $h_{\cdot,y+L} = h_{\cdot,y}$.*

The name comes from the fact that in a circulant matrix the $k$-th row is a circulant shift of the first row by $k-1$ steps to the right. That, is if $(J_0, J_1, \dots, J_{L-1})$ is the first row then the second is $(J_{L-1}, J_0, \dots, J_{L-2})$, the third $(J_{L-2}, J_{L-1}, J_0, \dots, J_{L-3})$ and altogether

$$\begin{pmatrix} J_0 & J_1 & J_2 & \dots & J_{L-1} \\ J_{L-1} & J_0 & J_1 & \dots & J_{L-2} \\ & & \ddots & & \\ J_2 & \dots & J_{L-1} & J_0 & J_1 \\ J_1 & \dots & J_{L-2} & J_{L-1} & J_0 \end{pmatrix} . \tag{81}$$

We see that it is enough to know the vector of (hopping) amplitudes $J_z = h_{1,1+z}$ for $z = 1, \dots, L-1$ to describe the whole matrix $h$. A translation invariant Hamiltonian $H(h)$ has a couplings matrix which is circulant but also Hermitian. This means that $J_{L-1} = J_1^*$, $J_{L-2} = J_2^*$ and in general $J_{L-z} = J_z^*$. In that case $J_0, J_1, \dots, J_{\lfloor L/2 \rfloor}$ are necessary to parametrize the matrix.

---

[1] $z$ could have smaller range but it doesn't harm

**Lemma 10** (Circulant matrices and discrete Fourier transforms). *A matrix $h$ is circulant if and only if it is diagonalized by a discrete Fourier transform. For $k = 1, \ldots, L$ the eigenvectors are*

$$\psi_k = \frac{1}{\sqrt{L}} \left( \phi_k, \phi_k^2, \ldots, \phi_k^{L-1}, 1 \right)^\top, \tag{82}$$

*where $\phi_k = e^{2\pi i k/L}$ and the corresponding eigenvalue read*

$$\lambda_k(h) = J_0 + \sum_{z=1}^{L-1} J_z e^{2\pi i z k/L} . \tag{83}$$

An important case is when the Hamiltonian couplings are real in addition to being circulant matrices and then we have

$$\lambda_k(h = h^\top = h^*) = J_0 + \sum_{z=1}^{\lfloor L/2 \rfloor} 2 J_z \cos(2\pi z k/L) . \tag{84}$$

In the most general translation invariant case, which is relevant for the case of conserved quantities if the initial covariance matrix was not purely real, we have

$$\lambda_k(h = h^\dagger) = J_0 + 2 \sum_{z=1}^{\lfloor L/2 \rfloor} \Re[J_z e^{2\pi i z k/L}] . \tag{85}$$

Here we have used translation invariance which in general reads $J_z = J_{L-z}^*$. As an example consider

$$\hat{I}_{z=1}(\eta = \pi/2) = \frac{1}{L} \sum_{x=1}^{L} (i \hat{f}_x^\dagger \hat{f}_{x+1} - i \hat{f}_{x+1}^\dagger \hat{f}_x) , \tag{86}$$

for which we have $\lambda_k = 2\Re[J_{z=1} e^{2\pi i z k/L}] = (2/L)\Re[i e^{2\pi i z k/L}] = -(2/L)\sin(2\pi z k/L)$.

*Proof.* To show the first direction, we will show that $\psi_{k'}^\dagger (h \psi_k) = \lambda_k(h) \delta_{k',k}$. We have

$$\psi_{k'}^\dagger (h \psi_k) = L^{-1} \sum_{x,y=1}^{L} h_{x,y} e^{\frac{2\pi i}{L}(ky - k'x)} \tag{87}$$

$$= L^{-1} \sum_{x,z=1}^{L} h_{x,x+z} e^{\frac{2\pi i}{L}(k-k')x} e^{\frac{2\pi i}{L}kz} \tag{88}$$

$$= L^{-1} \sum_{z=1}^{L} h_{1,1+z} e^{\frac{2\pi i}{L}kz} \sum_{x=1}^{L} e^{\frac{2\pi i}{L}(k-k')x} \tag{89}$$

$$= \sum_{z=1}^{L} h_{1,1+z} e^{\frac{2\pi i}{L}kz} \delta_{k,k'} , \tag{90}$$

which is by definition of $\lambda_k(h)$ what we were looking for. For the converse direction, we must show that a rotation by the discrete Fourier transform matrix of a spectrum $\lambda$ yields a circulant

matrix. We do this by checking the defining property

$$\tilde{h}_{x,y} = \left(\sum_{k=1}^{L} \lambda_k \psi_k \psi_k^\dagger\right)_{x,y} \tag{91}$$

$$= L^{-1} \sum_{k=1}^{L} \lambda_k e^{2\pi i k(x-y)/L} \tag{92}$$

$$= L^{-1} \sum_{k=1}^{L} \lambda_k e^{2\pi i k(x+z-y-z)/L} \tag{93}$$

$$= \tilde{h}_{x+z,y+z}. \tag{94}$$

Thus, the matrix $\tilde{h}$ is circulant. $\qquad\square$

### A.3 Bessel function asymptotics

A particularly insightful situation is the special case of a nearest-neighbour fermionic hopping Hamiltonian, setting $J_0 = 0$ and $J_1 = 1$. In this situation, we simply obtain

$$\omega_k = 2\cos(2\pi k/L), \tag{95}$$

and hence

$$G_{x,y}^*(t) = \frac{1}{L} \sum_{k=1}^{L} e^{-2i\cos(2\pi k/L)t + 2\pi i k(x-y)/L} \tag{96}$$

for the propagator. In the limit of large $L$, this can be seen as a Riemann sum approximation [54] to the integral

$$\tilde{G}_{x,y}^*(t) = \frac{1}{2\pi} \int_0^{2\pi} d\phi \, e^{-2i\cos(\phi)t} e^{i\phi(x-y)} = i^{x-y} \mathcal{J}_{x-y}(-2t), \tag{97}$$

where $\mathcal{J}_l : \mathbb{R} \to \mathbb{R}$ is the Bessel function of the first kind. The error in this approximation can be bounded from above as

$$|G_{x,y}^*(t) - i^{x-y} \mathcal{J}_{x-y}(-2t)| \leq \frac{\pi|x - y - 2t|}{L}. \tag{98}$$

These Bessel functions satisfy

$$|\mathcal{J}_{x-y}(-2t)| = O(t^{-1/2}) \tag{99}$$

for all $x, y$. That is to say, in this situation, one gets an equilibration following a $O(t^{-1/2})$ behaviour. This feature of the propagator is actually inherited by the actual correlation decay. In fact, a stronger statement can be made: $O(t^{-1/2})$ is not only an upper bound for $|\mathcal{J}_{x-y}(-2t)|$, but there cannot be a tighter uniform bound in the form of a power law. The asymptotics of Bessel functions [97] can be captured as

$$J_{x-y}(\tau) = \left(\frac{2}{\pi\tau}\right)^{1/2} \cos\left(\tau - \frac{(x-y)\pi}{2} - \frac{\pi}{4}\right) + O(|\tau|^{-1}), \tag{100}$$

for $\tau > 0$, showing that no tighter uniform power law bound can exist.

# B Steady states and local conservation laws via circulant matrices

Now we prove that dephasing under Hamiltonians that are only minimally degenerate leads to steady states with $\Gamma_{x,y}^{(eq)} \approx \langle \hat{I}_{|x-y|} \rangle$ where as in the main text we take the index arithmetic to be modulo $L$.

**Lemma 11** (Steady state covariance matrices as approximately circulant matrices). *Consider a state with covariance matrix $\Gamma$ and exponentially decaying correlations. For any $\hat{H}(h)$ with dispersion relation satisfying $\omega_k = \omega_{k'}$ only for $k = k'$, or $k = L - k'$ where $k' > k$ the steady state is approximately a circulant matrix with entries*

$$|\Gamma_{x,y}^{(eq)} - \langle \hat{I}_{|x-y|} \rangle| = O(L^{-1}) \,. \tag{101}$$

In particular, this holds true for the nearest-neighbour hopping model with dispersion relation $\omega_k = \cos(2\pi k / L)$.

*Proof.* Let us consider the action of the dephasing map on the initial covariance matrix $\Gamma^{(eq)} = \lim_{T \to \infty} \frac{1}{T} \int_0^T \Gamma(t)$ for two sites $x, y$ which reads

$$\Gamma_{x,y}^{(eq)} = \lim_{T \to \infty} \frac{1}{T} \int_0^T \Gamma_{x,y}(t) \tag{102}$$

$$= \lim_{T \to \infty} \frac{1}{T} \int_0^T \sum_{x',y'=1}^{L} G_{x,x'}(t) \Gamma_{x',y'} G_{y',y}^*(t) \tag{103}$$

$$= L^{-2} \sum_{x',y'=1}^{L} \Gamma_{x',y'} \sum_{k,k'=1}^{L} \left( \lim_{T \to \infty} \frac{1}{T} \int_0^T e^{(-i\omega_k t + i\omega_{k'})t} \right) e^{\frac{2\pi i}{L}(k(x-x')-k'(y-y'))} \tag{104}$$

$$= L^{-2} \sum_{x',y'=1}^{L} \Gamma_{x',y'} \sum_{k,k'=1}^{L} \delta_{\omega_k,\omega_{k'}} e^{\frac{2\pi i}{L}(k(x-x')-k'(y-y'))} \,. \tag{105}$$

Next, we will use the assumption concerning the minimal degeneracy of the dispersion relation which gives

$$\Gamma_{x,y}^{(eq)} = L^{-2} \sum_{x',y'=1}^{L} \Gamma_{x',y'} \sum_{k,k'=1}^{L} \delta_{k,k'} e^{\frac{2\pi i}{L}(k(x-x')-k'(y-y'))} \tag{106}$$

$$+ L^{-2} \sum_{x',y'=1}^{L} \Gamma_{x',y'} \sum_{k,k'=1}^{L} \delta_{k,L-k'}(1 - \delta_{k,k'}) e^{\frac{2\pi i}{L}(k(x-x')-k'(y-y'))} \,. \tag{107}$$

We notice that the condition $k = k'$ gives

$$\delta_{x-y,x'-y'} = L^{-1} \sum_{k=1}^{L} e^{\frac{2\pi i}{L}k(x-x'-y+y')} \,. \tag{108}$$

The other condition $k = L-k'$ also leads to simplification but one needs to be careful to observe that for $L$ even we may obtain $k = L-k'$ and $k = k' = L/2$ and such terms are already included in the previous sum. Additionally, $k' = L$ gives no solution to $k = L - k'$ and so together we

have

$$\Gamma_{x,y}^{(\text{eq})} = L^{-1} \sum_{x',y'=1}^{L} \Gamma_{x',y'} \delta_{x-y,x'-y'} \tag{109}$$

$$+ L^{-2} \sum_{x',y'=1}^{L} \Gamma_{x',y'} \sum_{k,k'=1}^{L-1} \delta_{k,L-k'} e^{\frac{2\pi i}{L}(k(x-x')-k'(y-y'))} \tag{110}$$

$$- L^{-2} \sum_{x',y'=1}^{L} \Gamma_{x',y'} \sum_{k,k'=1}^{L-1} \delta_{k,L-k'} \delta_{k,k'} e^{\frac{2\pi i}{L}(k(x-x')-k'(y-y'))} . \tag{111}$$

Here we identify the first line to give a current expectation value $\langle \hat{I}_{|x-y|} \rangle$. The inner sum in second line gives $L\delta_{x+y-x'-y'} - 1$ and the third is either 0 or can be bounded from above

$$\left| \Gamma_{x,y}^{(\text{eq})} - \langle \hat{I}_{|x-y|} \rangle \right| \leq L^{-1} \sum_{x',y'=1}^{L} |\Gamma_{x',y'}| \delta_{x+y-x'-y'} + 2L^{-2} \sum_{x',y'=1}^{L} |\Gamma_{x',y'}|. \tag{112}$$

Finally, we make use of the exponential decay of correlations $|\Gamma_{x,x+z}| \leq C_{\text{Clust}} e^{-z/\xi}$, obtaining

$$L^{-2} \sum_{x',y'=1}^{L} |\Gamma_{x',y'}| \leq L^{-2} \sum_{z=0}^{L} \sum_{x=1}^{L} |\Gamma_{x,x+z}| \leq L^{-1} C_{\text{Clust}} \sum_{z=0}^{\infty} e^{-z/\xi} \tag{113}$$

$$\leq \frac{C_{\text{Clust}}}{1 - e^{1/\xi}} L^{-1} . \tag{114}$$

Employing a similar bound for the first term we obtain

$$\left| \Gamma_{x,y}^{(\text{eq})} - \langle \hat{I}_{|x-y|} \rangle \right| \leq C_I L^{-1} . \tag{115}$$

$\square$

**Lemma 12** (Relevant currents). *Consider a state with covariance matrix $\Gamma$ and exponentially decaying correlations parametrized by the correlation length $\xi > 0$. Then for any time $t$ we have*

$$|\Gamma_{x,y}^{(d)}(t)| \leq C_{\text{Clust}} e^{-d/\xi} . \tag{116}$$

*Proof.* After a technical calculation using the definition of $\Gamma^{(d)}$ we find

$$\left| \Gamma_{x,y}^{(d)}(t) \right| = \left| \sum_{z,w=1}^{L} G_{x,w}(t) \Gamma_{z+d,z} \delta_{w,z+d} G_{y,z}^*(t) \right| \tag{117}$$

$$\leq \max_{z=1,\dots,L} \left| \Gamma_{z,z+d} \right| \left( \sum_{w=1}^{L} \left| G(t)_{x,w} \right|^2 \right)^{1/2} \left( \sum_{z=1}^{L} \left| G_{y,z}(t) \right|^2 \right)^{1/2} \tag{118}$$

$$= \max_{z=1,\dots,L} \left| \Gamma_{z,z+d} \right| \tag{119}$$

$$\leq C_{\text{Clust}} e^{-d/\xi}. \tag{120}$$

The second line follows from the inequality

$$| \langle v | A | w \rangle | \leq \|A\| \sqrt{\langle v|v \rangle} \sqrt{\langle w|w \rangle} , \tag{121}$$

where we are thinking of $\Gamma_{z,z+d} \delta_{w,z+d}$ as a matrix, with operator norm given by $\max_z |\Gamma_{z,z+d}|$. The third line follows because $G(t)$ is a unitary matrix, so its rows and columns are orthonormal vectors. In the last line, we have used the definition of exponentially decaying correlations. $\square$

## C  Bound on oscillatory sums of sequences with compact Fourier representation

We would like to prove a general bound on oscillatory sums of the type appearing in the main text where the phase sequence can be decomposed in a Fourier series with bounded number of harmonics. Specifically, we define a smooth phase function $\Phi_t : [0, 2\pi] \to \mathbb{R}$ of the form

$$\Phi_t(p) = dp + t \sum_{z=1}^{R} J_z \cos(zp + \alpha_z) \,, \tag{122}$$

where $t, d, J_1, \ldots, J_R, \alpha_1, \ldots \alpha_R \in \mathbb{R}$ with $J \neq 0$. It will be convenient to define

$$\Phi(p) = \sum_{z=1}^{R} J_z \cos(zp + \alpha_z) \,, \tag{123}$$

which plays, e.g., the role of a dispersion relation and we have $\Phi_t^{(\kappa)} = t\Phi^{(\kappa)}$ for all higher-order derivatives $\kappa > 1$. If we additionally define $p_k = 2\pi k/L$, then the sequence of interest will be

$$\varphi_k = \Phi_t(p_k) \,, \tag{124}$$

for $k = 1, \ldots, L$, where $L$ as in the main text stands for the system size. Note that our results become non-trivial for $L \geq t_0$ where $t_0$ is the relaxation time which dependents only on $J_z$. Physically, it is always given that $L$ is asymptotically large giving a uniform small parameter, so for system sizes of interest our requirements should be fulfilled. Mathematically all our statements remain correct by defining $[t_0, t_R] = \emptyset$ if $t_0 \geq t_R$ but it should be stressed that when $L$ is large enough we obtain a very non-trivial bound with $t_0 < t_R$.

Before we state our main theorem of this section, let us make the following definitions. We will use the Kusmin-Landau bound and the role of stationary points will be taken by the roots of $\Phi_t'$ denoted by

$$\mathcal{S}^{(1)} = \{p \in [0, 2\pi] \text{ s.t. } \Phi_t'(p) = 0\} \tag{125}$$

and the extremal points of the group velocity $\Phi_t'$

$$\mathcal{S}^{(2)} = \{p \in [0, 2\pi] \text{ s.t. } \Phi_t''(p) = 0\} \,. \tag{126}$$

The former set of points are exactly the points of vanishing group velocity while for the latter the band curvature vanishes. Let us make additionally the following definition useful for Taylor expansions around roots $r \in \mathcal{S} = \mathcal{S}^{(1)} \cup \mathcal{S}^{(2)}$. In general for $r \in \mathcal{S}$ we define

$$\kappa_r \geq 1 \tag{127}$$

to be the minimal integer such that for $r \in \mathcal{S}^{(a)}$ where $a = 1, 2$ the $(\kappa_r + a)$'th derivative does not vanish $|\Phi_t^{(\kappa_r + a)}(r)| \neq 0$. Additionally,

$$\kappa_0 = \max_{r \in \mathcal{S}} \kappa_r \tag{128}$$

will set the scaling of the final bound in the following theorem.

**Theorem 13** (Dephasing bound). *There exist a constant relaxation time $t_0$ and a recurrence time $t_R = \Theta(L)$ such that, for all $t \in [t_0, t_R]$ we obtain the bound*

$$\frac{1}{L} \left| \sum_{k=1}^{L} e^{i\varphi_k} \right| \leq C_\# t^{-\gamma} \,, \tag{129}$$

*where*

$$C_{\#} = 6(2R+1)\max\left\{1, \min_{r\in\mathcal{S}^{(2)}} \frac{\left|\Phi^{(\kappa_r+2)}(r)\right|}{2C_{max}^{(3)}\kappa_r!}, \max_{r\in\mathcal{S}^{(2)}} \frac{8\,(\kappa_r!)^2\,C_{max}^{(3)}}{\left|\Phi^{(\kappa_r+2)}(r)\right|^2}, \max_{r\in\mathcal{S}^{(1)}} \frac{4\,\kappa_r!}{\left|\Phi^{(\kappa_r+1)}(r)\right|}\right\} \quad (130)$$

*and $\gamma = 1/(3\kappa_0) > 1/(6R)$ are constants. We can take $\gamma = 1/3$, provided there are no repeated roots $\Phi_t''(p) = \Phi_t'''(p) = 0$ which holds true in the generic case.*

This theorem will for example allow us to bound

$$|G_{x,y}(t)| = \frac{1}{L}\left|\sum_{k=1}^{L} e^{i\omega_k t + 2\pi i k d/L}\right| \leq C_{\#} t^{-\alpha} \quad (131)$$

or

$$|f_n(t)| = \frac{1}{L}\left|\sum_{s=1}^{L} e^{i(\omega_{(s+n)}-\omega_s)t + 2\pi i s(x-y-d)/L}\right| \leq C_{\#}(\tfrac{n\pi}{L})t^{-\alpha}\,, \quad (132)$$

as a function of time and with constants expressed in the analytic properties of $\Phi_t$. The following lemma attributed to Kusmin and Landau [49] will be our key tool.

**Lemma 14** (Kusmin-Landau bound). *Suppose $(\varphi_n)_{n\in\{1,\dots,N\}}$ are real numbers and suppose the gaps $\delta_n = (\varphi_{n+1} - \varphi_n)$ for $n \in \{1, \dots, N-1\}$ are (i) increasing $\delta_n \geq \delta_{n-1}$ and (ii) each gap satisfies $\delta_n \in [\lambda, 2\pi - \lambda]$ with $\lambda > 0$. Then we have*

$$\left|\sum_{n=1}^{N} e^{i\varphi_n}\right| \leq \cot(\lambda/4) \leq \frac{2\pi}{\lambda}\,, \quad (133)$$

*where the second inequality follows from $\cos(x) \leq 1$ and $\sin(x) \geq 2x/\pi$ for $x \in [0, \pi/2]$.*

To apply Lemma 14, we need to understand the discrete Kusmin gaps defined by

$$\delta_k = \varphi_{k+1} - \varphi_k\,, \quad (134)$$

and show that they are separated from $0$ and $2\pi$ by some $\lambda > 0$ on a constant number of intervals where they are also monotonous. Because $\varphi_k = \Phi_t(p_k)$ we can use the mean value theorem obtaining

$$\delta_k = \frac{2\pi}{L}\Phi_t'(\tilde{p}_k) \quad (135)$$

for some $\tilde{p}_k \in [p_k, p_{k+1}]$ and $2\pi/L$ is the size of the interval to which we apply the theorem. We denote the summation domain by

$$\mathcal{D} = \{1, 2, \dots, L\} \quad (136)$$

and collect the corresponding $\tilde{p}_k$ points in

$$\mathcal{I} = \{\tilde{p}_k : k = 1, \dots, L-1\}\,. \quad (137)$$

Observe, that by the mean-value relation (135) the Kusmin gaps are monotonous on some $\mathcal{I}_r \subset \mathcal{I}$ if $\Phi_t'$ is monotonous on $\overline{\mathcal{I}}_r = \text{conv}(\mathcal{I}_r)$. By using the mapping $k \mapsto \tilde{p}_k$, we can define intervals in $\mathcal{D}$ associated to any subset $\mathcal{I}_r \subseteq \mathcal{I}$ defining $\mathcal{D}_r = \{k \in \mathcal{D} \text{ s.t. } \tilde{p}_k \in \mathcal{I}_r\}$. We want to divide $\mathcal{D}$ into intervals where $\delta$ are monotonous. After an elementary application of the triangle inequality, to each such region we want to apply the Kusmin-Landau bound. It is important to notice that their number is bounded above by the maximal coupling range $R$ according to the following lemma.

**Lemma 15** (Roots). *The function $\Omega : [0, 2\pi] \to \mathbb{C}$ defined by*

$$\Omega(p) = a_0 + \sum_{z=1}^{R} (a_z e^{ipz} + b_z e^{-ipz}) \,, \tag{138}$$

*where $a_j, b_j \in \mathbb{C}$ has at most $2R$ roots as long as $a \neq 0$ or $b \neq 0$.*

*Proof.* Define a complex polynomial $Y : \mathbb{C} \to \mathbb{C}$ by $Y(u) = a_0 u^R + \sum_{z=1}^{R} a_z u^{z+R} + \sum_{z=1}^{R} b_z u^{R-z})$ and observe that it is not identically zero and has degree at most $2R$ and hence at most $2R$ roots. Further, note that when restricted to the unit circle $S_1$ in the complex plane we have $Y(e^{ip}) = e^{iRp} \Omega(p)$ for any $p \in [0, 2\pi]$. From this we see that whenever $\Omega(p) = 0$ for some $p \in [0, 2\pi]$ then $u = e^{ip}$ is a root of $Y$ because the multiplicative prefactor $e^{ip}$ does not remove any roots. Thus the number of roots of $\Omega$ cannot exceed the number of roots of $Y$ which is upper bounded by $2R$. $\qquad\square$

**Corollary 16** (Number of roots of phase functions). *The phase function $\Phi_t$ and all its derivatives $\Phi_t'$, $\Phi_t''$ etc. have at most $2R$ roots.*

As we can easily check without loss of generality we can assume that $\Phi_t'(0) = \Phi_t'(2\pi) = 0$ and hence the interior between consecutive extremal points in $\mathcal{S} = \mathcal{S}^{(1)} \cup \mathcal{S}^{(2)}$ defines at most $4R+2$ intervals where $\Phi_t'(p)$ and $\Phi_t''(p)$ have a fixed sign. Specifically, we define these intervals as the points in $\mathcal{I}$ that lie between two consecutive roots from $\mathcal{S}$ and denote them by $\mathcal{I}_r \subset \mathcal{I}$ and note that $r$ ranges from 1 to some $R_0 \leq 4R + 2$. If, e.g., $\Phi_t(p) = \cos(p)$, then $R = 1$ and we have $R_0 = 4$ regions. We now establish condition *i)* of the Kusmin-Landau Lemma which concerns monotonicity.

**Lemma 17** (Monotonicity). *Let $r < r_2$ be two consecutive points belonging to $\mathcal{S}$. Then the Kusmin-Landau gaps $\delta_k$ are monotonous for all points $k$ corresponding to the interval $\mathcal{I}_r = \mathcal{I} \cap [r, r_2]$.*

*Proof.* Between $r$ and $r_2$ the first derivative of the phase function $\Phi_t'$ must be non-zero or otherwise there would be an intermediate root which is not possible as $r$ and $r_2$ are consecutive. There is also no intermediate root of the second derivative $\Phi_t''$ so it must have a fixed sign on the interior of the interval hence the derivative $\Phi_t'$ is either weakly increasing or decreasing and so the Kusmin-Landau gaps $\delta_k$ must be monotonous. $\qquad\square$

It will be useful to observe that any $\kappa_r$ can be bounded by the range $R$.

**Lemma 18** (Bounds from the range). *We have $\kappa_0 \leq 2R$.*

*Proof.* Consider again the non-zero polynomial $Y$ associated to $\Phi_t'$ or $\Phi_t''$ as described above. Then $Y$ has degree at most $\deg(Y) \leq 2R$ and because $J \neq 0$ we have that $\Phi_t \neq \texttt{const}$ and so $Y \neq 0$. Now, if we had that $Y(z_0) = Y'(z_0) = \ldots = Y^{(2R)}(z_0) = 0$ then, for any $z$ by Taylor expansion, we would find

$$Y(z) = \sum_{n=0}^{2R} \frac{Y^{(n)}(z_0)}{n!} (z - z_0)^n = 0 \,. \tag{139}$$

Thus, for $Y(e^{ip}) \neq 0$ to be true at some point $e^{ip}$ then $Y^{(n)}(z_0) \neq 0$ must be true for some $n \leq 2R$. $\qquad\square$

With this definition we can further set the constants

$$t_0 := \max\left\{1, \max_{r \in \mathcal{S}^{(2)}}\left|\frac{1}{\kappa_r + 1}\frac{\Phi^{(\kappa_r+3)}(r)}{\Phi^{(\kappa_r+2)}(r)}\right|^{3\kappa_r},\right.$$

$$\left.\max_{r \in \mathcal{S}^{(1)}}\left|\frac{1}{\kappa_r + 1}\frac{C_{\max}^{(\kappa_r+2)}}{\Phi^{(\kappa_r+1)}(r)}\right|^{3\kappa_r}, \left(\frac{C_0 + 1}{\min_{q,r \in \mathcal{S}}|q - r|}\right)^{2R+2}\right\} \quad (140)$$

and

$$t_{\mathrm{R}} = \frac{L}{4\max\{C_{\max}^{(1)}, C_1\}} . \quad (141)$$

This quantity is finite and independent of $L$ by the above remark and definition of $\kappa_r$, and because the numerator can be upper bounded by

$$C_{\max}^{(\kappa)} = \sum_{z=1}^{R}z^\kappa|J_z| \le R^{\kappa+1}\max_z|J_z| . \quad (142)$$

Furthermore, we define the time-independent constant

$$C_0 = \min_{r \in \mathcal{S}^{(2)}}\frac{\left|\Phi^{(\kappa_r+2)}(r)\right|}{2C_{\max}^{(3)}k_r!} . \quad (143)$$

We will now show that, after removing a small amount of points close to the border from each of the intervals $\mathcal{I}_r$, for the remaining points the Kusmin gaps will be lower and upper bounded. More precisely we define the two scalings that we shall use

$$p_t = C_0 t^{-1/3} \quad \text{and} \quad q_t = t^{-1/(3\kappa_r)} . \quad (144)$$

*Proof of Theorem 13.* Let us use the elementary observation that

$$\frac{1}{L}\left|\sum_{k=1}^{L}e^{i\varphi_k}\right| = \frac{1}{L}\left|\sum_{k=1}^{L}e^{i\varphi_{k+a}}\right| \quad (145)$$

for any $a$ together with the fact that our phase function is always periodic up to a constant

$$\Phi_t(p - r) = \Phi_t(p) - dr . \quad (146)$$

Observing that in the absolute value of the total sum any constant term in $\Phi_t$ drops out, we may assume that $\Phi_t'(0) = 0$ without loss of generality. Then using that the cosine functions are $2\pi$ periodic we also find that $\Phi_t'(2\pi) = 0$. With this step we reduced the total sum to a sum over the intervals $\mathcal{I}_r$ where the boundary points are appropriate roots.

Consider $r \in \mathcal{S}$ and the corresponding interval $\mathcal{I}_r$. Without loss of generality we may assume that $\Phi_t'(r) < \Phi_t'(r_2)$ and hence our task is to lower bound the Kusmin gaps around $r$. (If on $\mathcal{I}_r$ the gaps are negative then we can simply lower bound $\Phi_t'^{(-)} = -\Phi_t'$, while in the case $\Phi_t'(r) > \Phi_t'(r_2)$ we would have to lower bound the Kusmin gaps around $r_2$ which can be done the same way). By the monotonicity lemma, this assumption implies $\Phi_t'' > 0$ on $\mathcal{I}_r$.

**Step 1:** Restrict $\mathcal{I}_r$ to $\mathcal{I}_r \cap \mathcal{S}_t^c$ where

$$\mathcal{S}_t^c = [0, 2\pi]\backslash\{q \in [0, 2\pi) \text{ s.t. } |r - q| \le p_t + q_t \text{ for all } r \in \mathcal{S}\} , \quad (147)$$

such that $\delta_k \geq \lambda$ for

$$\lambda = \frac{2\pi C_1}{L} t^{1/3} \tag{148}$$

and

$$C_1 = \tfrac{1}{4} \min \left\{ \min_{r \in \mathcal{S}^{(1)}} \frac{\left|\Phi^{(\kappa_r+1)}(r)\right|}{\kappa_r!}, \min_{r \in \mathcal{S}^{(2)}} \frac{\left|\Phi^{(\kappa_r+2)}(r)\right|}{\kappa_r!} C_0 \right\} . \tag{149}$$

**Step 1, case 1:** $r \in \mathcal{S}^{(1)}$.
In this step, we expand around $r$, to obtain

$$\Phi'_t(r+q_t) = \frac{\Phi_t^{(\kappa_r+1)}(r)}{\kappa_r!} q_t^{\kappa_r} + \frac{\Phi_t^{(\kappa_r+2)}(\tilde{q})}{(\kappa_r+1)!} q_t^{\kappa_r+1}, \tag{150}$$

where the Lagrange error term in the first line is evaluated at some $\tilde{q} \in [r, r+q_t]$. We will show that for $t \geq t_0$ we have

$$\Phi_t^{(\kappa_r+1)}(r) \geq \frac{\Phi_t^{(\kappa_r+2)}(\tilde{q})}{\kappa_r+1} q_t , \tag{151}$$

which implies

$$\Phi'_t(r+q_t) \geq \frac{1}{2} \frac{\Phi_t^{(\kappa_r+1)}(r)}{\kappa_r!} q_t^{\kappa_r} . \tag{152}$$

We have that $\Phi'_t(p) > 0$ so by Eq. (150) we infer using (151) that $\Phi_t^{(\kappa_r+1)}(r) > 0$ and hence we have a non-trivial lower bound of the form

$$\Phi'_t(r+q_t) \geq \lambda_r = \frac{1}{2} \frac{\Phi_t^{(\kappa_r+1)}(r)}{\kappa_r!} t^{-1/3} = \frac{1}{2} \frac{\Phi^{(\kappa_r+1)}(r)}{\kappa_r!} t^{2/3} \tag{153}$$

and observe that $2\pi\lambda_r/L \geq \lambda$. It thus remains to show (151) which follows easily noticing that we can make $q_t$ sufficiently small using $t \geq t_0$. This condition is implied by finding that

$$\Phi^{(\kappa_r+1)}(r) \geq \frac{C_{\max}^{(\kappa_r+2)}}{\kappa_r+1} t^{-1/(3\kappa_r)} , \tag{154}$$

which is equivalent to

$$t \geq \left( \frac{1}{\kappa_r+1} \frac{C_{\max}^{(\kappa_r+2)}}{\Phi^{(\kappa_r+1)}(r)} \right)^{3\kappa_r} . \tag{155}$$

This can be shown to be true by invoking the definition of $t_0$.
**Step 1, case 2:** $r \in \mathcal{S}^{(2)}$.
Expanding around $r+q_t$ we obtain

$$\Phi'_t(r+q_t+p_t) = \Phi'_t(r+q_t) + \Phi''_t(r+q_t)p_t + \tfrac{1}{2}\Phi'''_t(\tilde{q})p_t^2, \tag{156}$$

where the Lagrange error term in the first line is evaluated at some $\tilde{q} \in [r+q_t, r+q_t+p_t]$. Note that we choose $q_t$ and $p_t$ small enough such that there is no repeated roots at this step. Because $\Phi'_t(r+q_t) \geq 0$

$$\Phi'_t(r+q_t+p_t) \geq \Phi'_t(r+p_t+q_t) - \Phi'_t(r+q_t) \tag{157}$$

$$\geq \Phi''_t(r+q_t)p_t + \tfrac{1}{2}\Phi'''_t(\tilde{q})p_t^2 . \tag{158}$$

We will show below that

$$\Phi_t''(r + q_t) \geq \Phi_t'''(\tilde{q}) p_t, \tag{159}$$

which directly implies

$$\Phi_t'(r + q_t + p_t) \geq \tfrac{1}{2}\Phi_t''(r + q_t) p_t . \tag{160}$$

We next continue to expand $\Phi_t''(r + q_t)$ around $r$ using the Taylor expansion

$$\Phi_t''(r + q_t) = \frac{\Phi_t^{(\kappa_r+2)}(r)}{\kappa_r!} q_t^{\kappa_r} + \frac{1}{2}\frac{\Phi_t^{(\kappa_r+3)}(\tilde{q})}{(\kappa_r + 1)!} q_t^{\kappa_r+1}, \tag{161}$$

where the last term is the Lagrange error term, so $\tilde{q} \in [r, r + q_t]$ and $\kappa_r \geq 1$ was defined above. We check that $q_t$ is sufficiently small such that $\left|\Phi_t^{(\kappa_r+2)}(r)\right| \geq \left|\frac{\Phi_t^{(\kappa_r+3)}(r)}{\kappa_r+1} q_t\right|$. Indeed, using $t \geq t_0$ leads to

$$q_t^{-1} = t^{1/(3\kappa_r)} \geq t_0^{1/(3\kappa_r)} \geq \frac{1}{\kappa_r + 1}\left|\frac{\Phi_t^{(\kappa_r+3)}(r)}{\Phi_t^{(\kappa_r+2)}(r)}\right|, \tag{162}$$

which after a simple rearrangement leads to that observation. This in turn implies that

$$\Phi_t''(r + q_t) \geq \frac{1}{2}\frac{\Phi_t^{(\kappa_r+2)}(r)}{\kappa_r!} q_t^{\kappa_r} . \tag{163}$$

Note that this is a non-trivial bound as due to the monotonicity on $\mathcal{I}_r$ we have $\Phi_t'' > 0$ and so $\Phi_t^{(\kappa_r+2)}(r)$ cannot be negative because the other term on the right hand side of (161) would be too small to make the whole right hand side positive. We are now in the position to check that condition (159) is satisfied which is implied by showing

$$\frac{1}{2}\frac{\Phi^{(\kappa_r+2)}(r)}{\kappa_r!} q_t^{\kappa_r} \geq C_{\max}^{(3)} p_t , \tag{164}$$

$$\frac{1}{2}\frac{\Phi^{(\kappa_r+2)}(r)}{\kappa_r!} t^{-1/3} \geq C_{\max}^{(3)} C_0 t^{-1/3} , \tag{165}$$

which is equivalent to

$$C_0 \leq \frac{\Phi^{(\kappa_r+2)}(r)}{2\kappa_r! C_{\max}^{(3)}} , \tag{166}$$

again using $\Phi_t^{(\kappa_r+2)}(r) \geq 0$ we find that this is true by comparing to the definition (143). With this result we obtain the lower bound (160) and explicitly inserting the time dependence arrive at

$$\Phi_t'(r + q_t + p_t) \geq \lambda_r = \frac{1}{4\kappa_r!}\Phi_t^{(\kappa_r+2)}(r) C_0 t^{-2/3} = \frac{1}{4\kappa_r!}\Phi^{(\kappa_r+2)}(r) C_0 t^{1/3} , \tag{167}$$

where again we find $2\pi\lambda_r/L \geq \lambda$, as desired.

**Step 1 summary:** Using (135) we obtain the following uniform bound lower bound $\delta_k \geq \lambda$ for $k \in \mathcal{I}_r \cap \mathcal{S}_t^c$ and any $r \in \mathcal{S}$.

**Step 2:** Upper bound $|\delta_k| \leq 2\pi - \lambda$. We show this by the bound

$$|\delta_k| \leq \frac{2\pi}{L} \max_{p \in [0, 2\pi)} \left|\Phi_t'(p)\right| \leq \frac{2\pi}{L}(t C_{\max}^{(1)} + |d|) . \tag{168}$$

Note that we can always take $|d| \leq L/2$. To see this, suppose that, e.g., $L > d = x - y > L/2$. Then we can replace $x$ by $x' = x + L$, which does not affect the propagator, but now we have $|x' - y| \leq L/2$. A similar trick works if $x - y < -L/2$. So we can upper bound $2\pi|d|/L$ by $\pi$, and we

$$|\delta_k| \leq \frac{2\pi t C_{\max}^{(1)}}{L} + \pi . \tag{169}$$

Next, we make use of Eq. (141) to see that

$$\frac{2\pi t C_{\max}^{(1)}}{L} + \lambda \leq \frac{2\pi t C_{\max}^{(1)}}{L} + \frac{2\pi}{L} C_1 t \leq \pi , \tag{170}$$

which implies

$$|\delta_k| \leq 2\pi - \lambda . \tag{171}$$

Hence for each $\mathcal{I}_r$ we can apply the Kusmin bound for times $t_0 \leq t \leq t_R$.

**Step 3:** Use the Kusmin-Landau lemma and obtain the final bound.

Summing up the discarded contribution and taking into account the bound on the number of the monotonous intervals we obtain the bound

$$\frac{1}{L}\left|\sum_{k=1}^{L} e^{i\varphi_k}\right| \leq (4R + 2)\left[\frac{p_t + q_t}{\pi} + C_1^{-1} t^{-1/3}\right] \leq C_{\#} t^{-1/(3\kappa_0)} , \tag{172}$$

where we have used that there are at most $4R+2$ Kusmin-Landau intervals that we restrict each at the edges by fewer than $L2(p_t + q_t)/\pi$ points and where the last term is the Kusmin-Landau bound. Inspecting the definition of $C_1$ we find that

$$C_1^{-1} = 4\max\left\{\max_{r \in \mathcal{S}^{(1)}} \frac{\kappa_r!}{\left|\Phi^{(\kappa_r+1)}(r)\right|}, 2C_{\max}^{(3)} \max_{r \in \mathcal{S}^{(2)}} \frac{(\kappa_r!)^2}{\left|\Phi^{(\kappa_r+2)}(r)\right|^2}\right\} . \tag{173}$$

Here, we have put the absolute values such that the bound in this form remains valid for monotonously growing and decreasing intervals. Hence, the constant $C_{\#}$ reads

$$C_{\#} := 6(2R + 1)\max\left\{1, \min_{r \in \mathcal{S}^{(2)}} \frac{\left|\Phi^{(\kappa_r+2)}(r)\right|}{2C_{\max}^{(3)} \kappa_r!}, \max_{r \in \mathcal{S}^{(2)}} \frac{8(\kappa_r!)^2 C_{\max}^{(3)}}{\left|\Phi^{(\kappa_r+2)}(r)\right|^2}, \max_{r \in \mathcal{S}^{(1)}} \frac{4\kappa_r!}{\left|\Phi^{(\kappa_r+1)}(r)\right|}\right\} . \tag{174}$$

**Generic case.** Let us finally remark on the generic case assuming there are no points for which $\Phi''(p) = \Phi'''(p) = 0$. For $r \in \mathcal{S}^{(1)}$ we can set $\kappa_r = 1$ whenver $\Phi''(r) \neq 0$. If $\Phi'(r) = \Phi''(r) = 0$ then in the generic case we will have $\Phi'''(r) \neq 0$ which would yield $\kappa_r = 2$ but then our bound would be dominated by $q_t = t^{-1/6}$ which we can improve. Instead expanding in $q_t$ we expand in $w_t = t^{-1/3}$ obtaining the equation

$$\Phi_t'(r + w_t) = \frac{\Phi_t'''(r)}{2} w_t^2 + \frac{\Phi_t^{(3)}(\tilde{q})}{6} w_t^3 . \tag{175}$$

As only the expansion length has changed we would find along the same arguments the lower bound

$$\Phi_t'(r + w_t) \geq \lambda_r = \frac{\Phi_t'''(r)}{4} w_t^2 = \frac{\Phi_t'''(r)}{4} t^{1/3} . \tag{176}$$

Therefore we are removing $\sim t^{-1/3}$ points and $\lambda^{-1} \sim t^{-1/3}$ also so the terms contributed from this case will have the scaling $\sim t^{-1/3}$. For $r \in \mathcal{S}^{(2)}$ we set $\kappa_r = 1$ and directly get the lower bound (148) also with the scaling $\sim t^{-1/3}$.

$\square$

In the main text, we have stated that $C_\#$ can in fact to be taken in a simpler form in the generic case where we have no points such that $\Phi_t''(r) = \Phi_t'''(r) = 0$. This means that $C_0 \leq 1$. If $\kappa_r = 1$ then nothing changes in our expansions. For $r \in \mathcal{S}^{(1)}$ also $\kappa_r = 2$ is possible. In this case, inspecting Eq. (175) we find that find that $\Phi_t'''(r)/4$ is the contribution to the $C_1$ constant instead of $\Phi_t''(r)/2$. This means that altogether we can define

$$M = \frac{1}{4} \min \left\{ \min_{r \in \mathcal{S}^{(1)}} \left| \Phi^{(\kappa_r+1)}(r) \right|, \min_{r \in \mathcal{S}^{(2)}} \left| \Phi^{(3)}(r) \right|^2 \right\}, \tag{177}$$

which leads us to the simplified constant

$$C_\# = 6(2R+1) \max \left\{ 1, \frac{8 C_{\max}^{(3)}}{M^2} \right\}. \tag{178}$$

Finally, it is worth mentioning that one can go beyond this setting by breaking up the gaps into those in the window $[\lambda, 2\pi - \lambda]$ and those in the window $[2\pi + \lambda, 4\pi - \lambda]$. Then we can apply the Kusmin bound to terms in each window separately. One can shift the gaps in $[2\pi + \lambda, 4\pi - \lambda]$ by making the substitution $a_n \mapsto a_n - 2\pi n$, which leads to $\delta_n \mapsto \delta_n - 2\pi$. Because we have only shifted $a_n$ by multiples of $2\pi$, this does not affect the exponential sum. After this shift, $\delta_n$ are in the interval $[\lambda, 2\pi - \lambda]$, and we can apply the Kusmin bound. This way, we would get bounds on equilibration valid for times after $t_R$. One could continue this process further with windows $[2n\pi + \lambda, 2(n+1)\pi - \lambda]$ for $n \in \mathbb{N}$, as long as the number of windows is small compared to $L$. It would be interesting to see if this patch-working of the Kusmin-Landau method for long times breaks down at the Poincare recurrence time which is much longer than the finite size revival time.

# D  Quasi-free ergodicity

For clarity we restate the theorem from the main text.

**Theorem 19** (Free fermionic ergodicity). *Let $t \mapsto G(t)$ be the propagator for a non-interacting translation invariant fermionic Hamiltonian $\hat{H}(h)$ which is off-diagonal on the one-dimensional real-space lattice. Then for all times $t$ between a relaxation time $t_0 = O(1)$ up until a recurrence time $t_R = \Theta(L)$ the propagator obeys*

$$|G_{x,y}(t)| \leq C t^{-\gamma}, \tag{179}$$

*where $C, \gamma > 0$ are constants. We can take $\gamma = 1/3$, provided there are no points $p$ such that $E''(p) = E'''(p) = 0$ which is true for generic models.*

*Proof of theorem: Quasi-free ergodicity.* As was explained in the main text we need to formulate a phase function such that it evaluates to the phases of the propagator. This is achieved by

$$\Phi_t(p) = dp + t \sum_{z=1}^{R} J_z \cos(zp) + J_0, \tag{180}$$

which evaluates to

$$\varphi_k = \Phi_t(p_k) = t \omega_k + 2\pi d k/L \tag{181}$$

for $p = 2\pi/L$ and $d = x - y$. By Theorem 13, we hence find the bound with $C_\#$ given in Eq. (174) being system size independent as the couplings are fixed. The relaxation and recurrence times $t_0$ and $t_R$ are given implicitly by the constraints in the proof of Theorem 13. The generic behaviour of the exponent $\gamma = 1/(3\kappa_0)$ is obtained for $\kappa_0 = 1$ which is attained at the wavefront of the nearest-neighbour hopping model [50]. $\qquad\square$

# E   Equilibration of the covariance matrix

In this section we will bound the deviations of the time evolved second moments $\Gamma(t)$ from the equilibrium covariance matrix $\Gamma^{(\mathrm{eq})}$ defined in Eq. (19) by a uniform real-space average of the local current densities.

**Proposition 20** (Equilibration of second moments)**.** *Consider a fermionic system with initially exponentially decaying correlations and non-resilient second moments $\Gamma$. Then there exist a constant relaxation time $t_0$ and a recurrence time $t_R = \Theta(L)$ such that, for all $t \in [t_0, t_R]$,*

$$\left| \Gamma_{x,y}(t) - \Gamma_{x,y}^{(\mathrm{eq})} \right| \leq C_\Gamma t^{-\gamma}, \tag{182}$$

*where $C_\Gamma, \gamma > 0$ are constants.*

*Proof.* Our goal is to bound how quickly $\Gamma_{x,y}(t)$, where $\Gamma(t) = G(t)\Gamma G(t)^\dagger$, relaxes towards the equilibrium values. First notice that these equal a real-space average where the value depends only on the separation $d = \min\{|x-y|, |x-y+L|, |x-y-L|\}$. Let us define the decomposition of $\Gamma$ into its currents, that is $\Gamma = \sum_{d=-\lfloor (L+1)/2 \rfloor + 1}^{\lfloor L/2 \rfloor} \Gamma^{(d)}$ with entries

$$\Gamma_{x,y}^{(d)} = \Gamma_{x,y} \delta_{x,y+d}, \tag{183}$$

where we use the convention $\delta_{a,b+L} = \delta_{a,b}$. The evolution is linear, so

$$\Gamma(t) = \sum_{d=-\lfloor (L-1)/2 \rfloor + 1}^{\lfloor L/2 \rfloor} \Gamma^{(d)}(t), \tag{184}$$

where we define

$$\Gamma_{x,y}^{(d)}(t) := \left( G(t)\Gamma^{(d)}G(t)^\dagger \right)_{x,y} = \sum_{z,w}^{L} G_{x,w}(t)\Gamma_{w,z}^{(d)}G_{y,z}^*(t) \tag{185}$$

$$= \sum_{z,w=1}^{L} G_{x,w}(t)\Gamma_{w,z}\delta_{w,z+d}G_{y,z}^*(t) \tag{186}$$

$$= \sum_{z=1}^{L} G_{x,z+d}(t)\Gamma_{z+d,z}G_{y,z}^*(t). \tag{187}$$

Our target equilibrium ensemble has matrix elements given by

$$\Gamma_{x,y}^{(\mathrm{eq})} = \sum_{d=-\lfloor (L+1)/2 \rfloor + 1}^{\lfloor L/2 \rfloor} I_d \delta_{x,y+d}, \tag{188}$$

where specifically the value equilibrium values read

$$I_d = \frac{1}{L} \sum_x \Gamma_{x,x+d}. \tag{189}$$

If the initial covariance matrix is real then this is exactly the $d$-th current in the initial state. Otherwise, one has to consider also the 'complex' currents as discussed above. With these definitions, we obtain the bound

$$\left| \Gamma_{x,y}(t) - \Gamma_{x,y}^{(\mathrm{eq})} \right| \leq \sum_{d=-\lfloor (L+1)/2 \rfloor + 1}^{\lfloor L/2 \rfloor} \left| \Gamma_{x,y}^{(d)}(t) - I_d \delta_{x,y+d} \right| \tag{190}$$

by using the triangle inequality. After these steps organizing the notation, we will present a first non-trivial bound showing that in the above sum only the currents with $d \leq d_\xi(t)$ contribute significantly. This is natural because of the exponentially decaying correlations so denoting the correlation length as $\xi$ it suffices to use Lemma 12 with

$$d_\xi(t) = \xi \ln(t^{1/(3\kappa_0)}) , \tag{191}$$

where $\kappa_0$ is a positive constant which is indpendent of the system size and will be defined below. Then the currents $d > d_\xi(t)$ will negligibly contribute to $\|\Gamma(t) - \Gamma^{(\mathrm{eq})}\|_{\max}$ for sufficiently large $t > t_0$. So we consider $d \leq d_\xi(t)$. Now we expand $\Gamma^{(d)}$ via the discrete Fourier transform

$$\Gamma_{z+d,z} = \sum_{n=1}^{L} \mathcal{X}_n^{(d)} e^{2\pi i n z / L} . \tag{192}$$

Then we have that

$$\Gamma_{x,y}^{(d)}(t) = \sum_{z=1}^{L} G_{x,z+d}(t) G_{y,z}^*(t) \Gamma_{z+d,z} = \sum_{n=1}^{L} \mathcal{X}_n^{(d)} \sum_{z=1}^{L} G_{x,z+d}(t) G_{y,z}^*(t) e^{2\pi i n z / L} . \tag{193}$$

Recall the definition of the propagator

$$G_{x,y}(t) = \frac{1}{L} \sum_{k=1}^{L} \exp(i\omega_k t + 2\pi i k (x-y)/L) , \tag{194}$$

by which we get

$$\Gamma_{x,y}^{(d)}(t) = \frac{1}{L^2} \sum_{n=1}^{L} \mathcal{X}_n^{(d)} \sum_{r,s=1}^{L} \sum_{z=1}^{L} e^{i\omega_r t + 2\pi i r (x-z-d)/L} e^{-i\omega_s t - 2\pi i s (y-z)/L} e^{2\pi i n z / L} \tag{195}$$

$$= \frac{1}{L^2} \sum_{n=1}^{L} \mathcal{X}_n^{(d)} \sum_{r,s=1}^{L} e^{i(\omega_r - \omega_s)t + 2\pi i (rx - sy - rd)/L} \sum_{z=1}^{L} e^{2\pi i z(-r+s+n)/L} . \tag{196}$$

Next, we use

$$\sum_{z=1}^{L} e^{2\pi i z(-r+s+n)/L} = L \sum_{\mu \in \mathbb{Z}} \delta_{-r+s+n,\mu L} , \tag{197}$$

applying it to the sum over $r$ while summing over $s, n$. Then we find that $-r + s + n = \mu L$ has solutions with either $\mu = 0$ or $\mu = 1$ but not at the same time because of the variable range $r, s, n \in [L]$. Indeed, we always have $2 \leq s + n \leq 2L$ and so we have the unique solutions $r = s + n$ for $s + n \leq L$ and $r = s + n - L$ for $s + n \geq L$. Thus, using $\omega_{k+\mu L} = \omega_k$ which follows by inspecting the definition in Eq. (84) we get

$$\Gamma_{x,y}^{(d)}(t) = \frac{1}{L} \sum_{n=1}^{L} \mathcal{X}_n^{(d)} e^{2\pi i n (x-d)/L} \sum_{s=1}^{L} e^{i(\omega_{(s+n)} - \omega_s)t + 2\pi i s (x-y-d)/L} \tag{198}$$

$$= \sum_{n=1}^{L} \mathcal{X}_n^{(d)} e^{2\pi i n (x-d)/L} f_n(t) . \tag{199}$$

In the last line, we have defined

$$f_n(t) := \frac{1}{L} \sum_{s=1}^{L} e^{i(\omega_{(s+n)} - \omega_s)t + 2\pi i s (x-y-d)/L} . \tag{200}$$

The equilibrium currents will be uniform so we need to bound

$$\left|\Gamma_{x,y}^{(d)}(t) - I_d \delta_{x,y+d}\right| = \left|\sum_{n=1}^{L-1} \mathcal{X}_n^{(d)} e^{2\pi i n z/L} f_n(t)\right| , \tag{201}$$

because

$$I_d = \mathcal{X}_L^{(d)} . \tag{202}$$

As we have observed in the main text, we have

$$\omega_{(k+n)} - \omega_k = \sum_{z=1}^{R} K_z \sin\left(\frac{2\pi z k}{L} + \frac{\pi n z}{L}\right) , \tag{203}$$

with $K_z = -4 J_z \sin(\pi z n/L)$. In order to use the dephasing bound from theorem 13, we define

$$\Phi_t(p) = -4t \sum_{z=1}^{R} J_z \sin(\alpha z) \sin(z p + z \alpha) + p(x - y - d) \tag{204}$$

and by evaluating with $\alpha = \pi n/L$ and $p_k = 2\pi k/L$, we have

$$\varphi_k = \Phi_t(p_k) = (\omega_{(k+n)} - \omega_k) t + 2\pi i k (x - y - d)/L . \tag{205}$$

We hence obtain the bound

$$|f_n(t)| \le C_\#(\alpha) t^{-1/(3\kappa_0)} , \tag{206}$$

where now $C_\#$ depends on the derivatives of (204) evaluated at roots and we indicate the dependance on $\alpha$ as for $\alpha \approx 0$ the constant would not be system size independent. As long as $K_z$ have no stray dependence on $L$ these values are constant numbers in the system size so we can scale up the system size and get a non-trivial bound. All this is secured by using the assumption of non-resilient correlations which leads to

$$\left|\Gamma_{x,y}^{(d)}(t) - I_d \delta_{x,y+d}\right| = \sum_{\substack{n=1 \\ n\pi/L \in \mathcal{R}}}^{L-1} \left|\mathcal{X}_n^{(d)}\right| + \sum_{\substack{n=1 \\ n\pi/L \notin \mathcal{R}}}^{L-1} \left|\mathcal{X}_n^{(d)}\right| |f_n(t)| \tag{207}$$

$$\le C_{\mathrm{RS}} L^{-1} + C_{\mathrm{NRS}} C_{\mathrm{th}} t^{-1/(3\kappa_0)} \tag{208}$$

$$\le (C_{\mathrm{RS}} + C_{\mathrm{NRS}} C_{\mathrm{th}}) t^{-1/(3\kappa_0)} , \tag{209}$$

where we have used $L^{-1} \le t^{-1/(3\kappa_0)}$, which holds true for $t \le t_R = \Theta(L)$. We now finalize the total bound by

$$\left|\Gamma_{x,y}(t) - \Gamma_{x,y}^{(\mathrm{eq})}\right| \le 2 d_\xi(t) \max_{|d| \le d_\xi(t)} \left|\Gamma_{x,y}^{(d)}(t) - \Gamma_{x,y}^{(\mathrm{eq})}\right| + \frac{C_{\mathrm{Clust}}}{1 + e^{-1/\xi}} t^{-1/3\kappa_0} \tag{210}$$

$$\le \tfrac{1}{2} C_\Gamma \ln(t^{1/(3\kappa_0)}) t^{-1/(3\kappa_0)} + \frac{C_{\mathrm{Clust}}}{1 + e^{-1/\xi}} t^{-1/3\kappa_0} , \tag{211}$$

where we have defined

$$C_\Gamma := \max\left\{4\xi(C_{\mathrm{RS}} + C_{\mathrm{NRS}} C_{\mathrm{th}}), \frac{2 C_{\mathrm{Clust}}}{1 + e^{-1/\xi}}\right\} . \tag{212}$$

Observe that $\kappa_0 \le 2R$. Thus, for sufficiently large $t$ we have for some $\varepsilon > 0$ the final bound

$$\|\Gamma(t) - \Gamma^{(\mathrm{eq})}\|_{\max} = \max_{x,y} \left|\Gamma_{x,y}(t) - \Gamma_{x,y}^{(\mathrm{eq})}\right| \le C_\Gamma t^{-1/(3\kappa_0)+\varepsilon} . \tag{213}$$

$\square$

# F Examples of non-resilient second moments

## F.1 m-step periodic states

Suppose $\Gamma_{z+d,z}$ is $m$-step periodic, with $\ell = L/m \in \mathbb{N}$, so that $\Gamma_{z+d+m,z+m} = \Gamma_{z+d,z}$. We get

$$\mathcal{X}_n^{(d)} = L^{-1} \sum_{z=0}^{L-1} \Gamma_{z+d,z} e^{-2\pi i n z/L} \tag{214}$$

$$= L^{-1} \sum_{u=0}^{m-1} \sum_{v=0}^{L/m-1} \Gamma_{u+vm+d,u+vm} e^{-2\pi i n(u+vm)/L} \tag{215}$$

$$= \left( \frac{1}{m} \sum_{u=0}^{m-1} \Gamma_{u+d,u} e^{-2\pi i n u/L} \right) \left( \frac{m}{L} \sum_{v=0}^{L/m-1} e^{-2\pi i n v m/L} \right) \tag{216}$$

$$= \left( \frac{1}{m} \sum_{u=0}^{m-1} \Gamma_{u+d,u} e^{-2\pi i n u L} \right) \left( \sum_{\alpha=0}^{m-1} \delta_{n,\alpha\ell} \right) . \tag{217}$$

So all $\mathcal{X}_n^{(d)}$ are vanishing, except those with with $n = \alpha\ell$, where $\alpha \in \{0, \ldots, m-1\}$.

## F.2 Random dislocations

Suppose $\Gamma^{(d)}$ can be decomposed as $\Gamma^{(d)} = \Gamma^{(d,NR)} + \Gamma^{(d,SR)}$ where $\Gamma^{(d,NR)}$ is non-resilient and $\Gamma^{(d,NR)}$ has sparse support. Then $\Gamma^{(d)}$ is again non-resilient. This follows trivially as for the sparse part the Fourier transform is bounded by the inverse system size

$$|\mathcal{X}_n^{(d)}| \le L^{-1} \sum_{z=0}^{L-1} |\Gamma_{z+d,z}^{(d,SR)}| \le \frac{S}{L} , \tag{218}$$

where $S = O(1)$ is the number of the sparse entries in $\Gamma^{(d,SR)}$.

## F.3 Uniformly random currents

Take $\Gamma_{z+d,z} \in [a,b]$ to be uniformly and independently distributed. Then $\Gamma^{(d)}$ is non-resilient. Indeed, we find that on average, we have

$$\mathbb{E}[\mathcal{X}_n^{(d)}] = \frac{1}{L} \sum_{z=1}^{L} \mathbb{E}[\Gamma_{z+d,z}] e^{-2\pi i n z/L} \tag{219}$$

$$= \frac{a+b}{2} L^{-1} \sum_{z=0}^{L-1} e^{-2\pi i n z/L} \tag{220}$$

$$= \frac{a+b}{2} \delta_{n,L} . \tag{221}$$

We furthermore calculate the second moment using

$$\mathbb{E}[\Gamma_{x,y}^2] = \frac{a^2 + ab + b^2}{3} \tag{222}$$

to get

$$\mathbb{E}[|\mathcal{X}_n^{(d)}|^2] = \frac{1}{L^2} \sum_{z,s=1}^{L} \mathbb{E}[\Gamma_{z+d,z}\Gamma_{s+d,s}]e^{-2\pi i n(s-z)/L} \tag{223}$$

$$= \frac{1}{L^2} \sum_{\substack{z,s=1 \\ z \neq s}}^{L} \mathbb{E}[\Gamma_{z+d,z}\Gamma_{s+d,s}]e^{-2\pi i n(s-z)/L} + \frac{1}{L^2}\sum_{z=1}^{L} \mathbb{E}[\Gamma_{z+d,z}^2] \tag{224}$$

$$= \frac{a^2+2ab+b^2}{4L^2}\left(\sum_{z,s=1}^{L} e^{-2\pi i n(s+z)/L} - L\right) + \frac{a^2+ab+b^2}{3L} \tag{225}$$

$$= \mathbb{E}[\mathcal{X}_n^{(d)}]^2 + \frac{(a-b)^2}{12L} \,, \tag{226}$$

and hence the variance reads

$$\mathrm{Var}[\mathcal{X}_n^{(d)}] = \frac{(a-b)^2}{12L} \,. \tag{227}$$

By Chebyshev's inequality

$$\mathbb{P}\left(\left|\mathcal{X}_n^{(d)} - \mathbb{E}[\mathcal{X}_n^{(d)}]\right| \geq K\right) \leq \frac{\mathrm{Var}[\mathcal{X}_n^{(d)}]}{K^2} \,, \tag{228}$$

we obtain that

$$\left|\mathcal{X}_n^{(d)} - \mathbb{E}[\mathcal{X}_n^{(d)}]\right| \leq KL^{-1/2} \,, \tag{229}$$

with probability greater than $1-(CK)^{-2}$.

### F.4 Resilient example: $P = L/2$ - periodic block calculation

Say $L$ is even and we have a state with

$$\langle \hat{f}_x^{\dagger}\hat{f}_x \rangle = \begin{cases} 1, & x < L/2 \\ 0, & x \geq L/2 \,. \end{cases} \tag{230}$$

Then all currents with $d \neq 0$ vanish and we should calculate the Fourier transform of the diagonal of the covariance matrix namely

$$\mathcal{X}_n^{(d)} = \sum_{k=1}^{L/2-1} e^{2\pi i n k/L} = \begin{cases} \frac{1-e^{i\pi n}}{1-e^{i\pi}} = 1, & n \text{ is odd} \\ 0, & n \text{ is even} \,. \end{cases} \tag{231}$$

Therefore there is no chance that we can get a non-trivial bound of the form

$$\left|\Gamma_{x,y}(t) - \Gamma_{x,y}^{(\mathrm{eq})}\right| \leq \sum_{n=1}^{L}\left|\mathcal{X}_n^{(d)}\right||f_n(t)| \,, \tag{232}$$

because it will scale with the system size as $\sim LC_{\mathrm{th}}t^{-\gamma}/2$ due to the number of non-trivial harmonics.

## F.5 Details of quenches from disordered to translation invariant models

In this section, we provide more details on the discussion of the simularity of averaged generalized Gibbs ensembles with thermal ensembles. In this context, it is useful to discuss the exact quench state $\Gamma^{(\text{Quench})}(t)$ and the infinite-time average $\Gamma^{(\infty)}$ on the level of quasiparticle occupation numbers in momentum space. For the quenched state these stay constant for all times due to unitarity and we have checked that typically there are initially fluctuations around the idealized Fermi-Dirac distribution but a Fermi edge can be observed. However, in order to obtain the infinite time average we apply a dephasing channel which as it is not unitary does change the occupation numbers despite conserving the relevant local currents. In that case we have noticed a much smoother quasiparticle number distribution, resembling much closer the thermal Fermi-Dirac distribution. Additionally we noticed that for the same model with the noise uniformly distributed in $[0, w]$ the resulting equilibrium state is not thermal, but additionally considering a chemical potential leads to agreement. These observations suggest that the equilibration process that we have discussed analytically leads to a thermal steady state for the particular selection of initial states discussed here which are thermal states of a Hamiltonian with the same tunnelling range as the quench Hamiltonian. It appears to be an interesting question whether we can in general expect GGEs which are simply thermal steady-states in the natural case occurring in many physical instances where the kinetic energy is an inherent property of the system over which we have little control and we prepare thermal initial states by controlling only the on-site potential by external forces.

A peculiarity stemming from quasi-free integrability is that it is enough to have access to only one translation invariant quench Hamiltonian to prepare thermal states of any other translation invariant model just by assigning the initial correlation content. In particular being able to tune the correlation length is a crucial ingredient, but the initial correlations need not be translation invariant or even Gaussian, as we have discussed above. Thanks to the Gaussification result, one can also use many-body interactions to tune close to a phase transition in order to increase the correlation length even if the state obtained will be non-Gaussian. Indeed we have a proof of equilibration to a Gaussian state but now the steady state may acquire an unusually large correlation length for a thermal state of the quench Hamiltonian. This "one to rule them all" result shows that the properties of the equilibrated state may be unrelated to the range of the dynamics, which is slightly at odds with the usual approach to inferring microscopic properties of various materials. It would be interesting to see whether experiments measuring only conductivity or other linear response properties could be adversarily tricked to indicate always different dynamical models by getting different states as input while the true Hamiltonian is always merely the nearest neighbour model. Such interactive experiments may be possible with existing quantum simulation technologies [85]. On the other hand if precise microscopic measurements are limited, then observing just the fundamental qualitative properties such as the formation of the Fermi edge which determines solid state properties should be a generic feature independent of the memory effect due to integrability. As there is only few trailblazing works concerning what happens to a GGE in the presence of weak interactions [91–94], it would be exciting to study this systematically in optical lattices experimentally.

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
