# Peer review of "Equilibration towards generalized Gibbs ensembles in non-interacting theories"

_SciPost Physics, doi:SciPost Phys. 7, 038 (2019)_

## Round 2 · Referee Report · Anonymous (Referee 1) · 2019-8-17

Strengths

  1. Mathematical rigor.
  2. Generality.
  3. Breadth of interest.

Weaknesses

  1. Details of the proofs can be tedious to go through, but this is unavoidable in a rigorous treatment.

Report

This paper treats the approach to a state of thermal equilibrium, as described by a generalized Gibbs ensemble, for a large class of initial states of a system of noninteracting lattice fermions. It unifies, under a general formalism, a number of other results in the literature, and provides an overview for the general mechanism of thermalization. Therefore this work should be of interest to physicists across a number of disciplines.

In my judgment, this is a very high quality paper that deserves a wide audience.

Requested changes

None.

---

## Round 2 · Referee Report · Anonymous (Referee 2) · 2019-8-26

Strengths

  1. Exact and new results for local thermalization (in the sense of a generalized Gibbs ensemble), especially regarding the equilibration timescales
  2. Very well written paper

Weaknesses

  1. The results are only valid for non-interacting models, therefore there is no thermalization process that could lead to thermal mode occupation.

Report

In this paper the authors show in a mathematically exact manner how local equilibration occurs in translation-invariant non-interacting models. It is especially noteworthy that a timescale for local equilibration can be extracted (power law behavior), which is often missing in other discussions of equilibration in the literature (as the authors correctly point out).

The main limitation of this paper is that non-interacting models can only thermalize locally, but not in momentum space. The authors point this out as well in Eq. (16) where the momentum occupation numbers are conserved quantities. In solid state physics the momentum occupation numbers are easily accessible observables, for example via ARPES: The main question for thermalization of translation invariant systems is how such mode occupations become thermal. This question can obviously not be addressed in the setting of this paper. So the results in this paper are mathematically nice and important, but do not contribute to the main physical question.

Still this is a very well written paper with exact results, which should certainly be published.
  • validity: top
  • significance: good
  • originality: good
  • clarity: top
  • formatting: excellent
  • grammar: excellent

Author:  Marek Gluza  on 2019-10-07  [id 621]

(in reply to Report 2 on 2019-08-26)
Category:
remark
suggestion for further work

We thank the Referee very much for his or her positive comments.

We agree that systems which can be studied with ARPES are outside the scope of our work as we are treating only the limiting cases where e.g. phonons are effectively uncoupled from electrons and there is negligible many-body scattering. Our theory could however be interesting for experiments in optical lattices where one could systematically tune the interactions.

---

## Editorial Decision

published